# The Genus *Cladosporium*: A Prospective Producer of Natural Products

**DOI:** 10.3390/ijms25031652

**Published:** 2024-01-29

**Authors:** Yanjing Li, Yifei Wang, Han Wang, Ting Shi, Bo Wang

**Affiliations:** 1College of Chemical and Biological Engineering, Shandong University of Science and Technology, Qingdao 266590, China; 15153232290@163.com (Y.L.); kingsley11f115@163.com (Y.W.); h15725209196@163.com (H.W.); 2State Key Laboratory of Microbial Technology, Institute of Microbial Technology, Shandong University, Qingdao 266200, China

**Keywords:** *Cladosporium* sp., marine fungi, biological activity, medicinal potentiality, secondary metabolites

## Abstract

*Cladosporium*, a genus of ascomycete fungi in the *Dematiaceae* family, is primarily recognized as a widespread environmental saprotrophic fungus or plant endophyte. Further research has shown that the genus is distributed in various environments, particularly in marine ecosystems, such as coral reefs, mangroves and the polar region. *Cladosporium*, especially the marine-derived *Cladosporium*, is a highly resourceful group of fungi whose natural products have garnered attention due to their diverse chemical structures and biological activities, as well as their potential as sources of novel leads to compounds for drug production. This review covers the sources, distribution, bioactivities, biosynthesis and structural characteristics of compounds isolated from *Cladosporium* in the period between January 2000 and December 2022, and conducts a comparative analysis of the *Cladosporium* isolated compounds derived from marine and terrestrial sources. Our results reveal that 34% of *Cladosporium*-derived natural products are reported for the first time. And 71.79% of the first reported compounds were isolated from marine-derived *Cladosporium*. *Cladosporium*-derived compounds exhibit diverse skeletal chemical structures, concentrating in the categories of polyketides (48.47%), alkaloids (19.21%), steroids and terpenoids (17.03%). Over half of the natural products isolated from *Cladosporium* have been found to have various biological activities, including cytotoxic, antibacterial, antiviral, antifungal and enzyme-inhibitory activities. These findings testify to the tremendous potential of *Cladosporium*, especially the marine-derived *Cladosporium*, to yield novel bioactive natural products, providing a structural foundation for the development of new drugs.

## 1. Introduction

The fungus *Cladosporium* sp. belongs to the *Dematiaceae* family in the *Dothideomycetes* order of *Ascomycota* [1]. It was initially classified by Link in 1815 [2]. The mycelium of the colonies is well-developed, branched and colored white at the edges. It appears floccose, with a dark center and a white mycelium coating, gradually fading to white margins. In the center, the mycelium is dark and raised, containing conidia in varying shades of olive, along with clear spore scars and umbilicals [3,4,5]. The three primary species of the fungi are *C. herbarum*, *C. cladosporioides* and *C. sphaerospermum* [6]. It includes many important phytopathogenic bacteria that cause stem rot and leaf spot diseases, such as yellow rot, which is the causal agent of leaf mold in tomato [7]. Several species are commonly found as contaminants in clinical laboratories and can prompt allergic lung diseases [8,9]. For instance, *C. sphaerospermum*, a predominantly indoor fungus, has been reported in cases of meningitis, subcutaneous and intra-bronchial infections [8]. Besides, *C. cladosporioides* is a dark mold that can be found on outdoor and indoor materials worldwide. It is one of the most common fungi in outdoor air, and its spores are important in seasonal allergies. While it rarely causes invasive diseases in animals, it is a significant pathogen for plant diseases, attacking the leaves and fruits of many plants [9]. However, some plants have been found to benefit from *Cladosporium*. For example, *C. sphaerospermum*, isolated from glycine roots, promotes plant growth [10]. Some species of genus *Cladosporium* are also associated with allergic rhinitis, respiratory arrest in asthmatics and brown filariasis in humans and animals [11,12].

In 2000, cladosporide A was found as a characteristic antifungal agent against the human pathogenic filamentous fungus *Aspergillus fumigatus* from *Cladosporium* sp. IFM49189 by Tomoo Hosoe et al. [13], which is the first natural product discovered from the genus *Cladosporium*. Although the first research on the secondary metabolites of the genus *Cladosporium* in 2000 occurred more than one century after the first identification of the genus *Cladosporium* in 1815, the discovery of this characteristic antifungal agent opened the door to the study of natural products from *Cladosporium*. Some species produce compounds with insecticidal activity that may have potential as biological controls in agriculture and forestry [14,15,16]. *Cladosporium* are abundantly parasitized in soil, sea and plant bodies, and in addition to synthesizing compounds consistent with or similar to those of their hosts, they also synthesize structurally diverse compounds such as polyketides and alkaloids, macrocyclic endolipids, steroids and terpenoids, which have potent or moderate biological activities [17,18], making them potentially useful in the treatment of diseases and agricultural applications.

Marine organisms, particularly fungi-derived compounds display a variety of biological activities, including cytotoxicity [19], antibacterial [20], antiviral [21], antifungal [22] and enzyme-inhibitory activities [23], which can produce a wide range of bioactive secondary metabolites with possible pharmaceutical applications. Consequently, these compounds have emerged as promising candidates for addressing the treatment of challenging diseases such as human colon cancer, Hepatitis B Virus (HBV) and inflammation [24]. Based on the investigation’s findings, we discovered that *Cladosporium* from the ocean exhibited a greater abundance of secondary metabolites. Marine-derived *Cladosporium* are a valuable source of diverse compounds that have been extensively studied for their potential biomedical applications. They show a diversity of structures, including polyketides [25], alkaloids [26], steroids and terpenoids [21], benzene derivatives [27] as well as cyclic peptides [28]. The rich variety of secondary metabolite chemicals found in marine organisms can be attributed to several factors. The marine environment is unique in terms of pH, temperature, pressure, oxygen, light and salinity. Moreover, it also occupies most of the earth’s surface and nearly 87% of the world’s biosphere, the complexity and diversity of the marine environment offer abundant living space and resources for microorganisms, leading to increased competition and interaction among them. Additionally, interactions between microorganisms in the ocean and other organisms, such as symbiotic relationships with marine plants or animals, can also influence the diversity of secondary metabolites as they may lead to the production of specific compounds by microorganisms [27].

Although the secondary metabolites of *Cladosporium* have been reviewed [29,30,31], the analysis of possible biosynthetic pathways, multi-dimensional comparative analysis of natural products from marine and terrestrial sources, structure–activity relationships, and activity-based molecular network analysis have not been conducted. This review, through these analysis methods, aimed to explore the research on the secondary metabolites, especially the differences between marine- and terrestrial-derived compounds, of genus *Cladosporium*, including their chemical structures and biological activities, as well as their biosynthetic pathways. Herein, this work surveyed all the studies about the natural products isolated from *Cladosporium* published between January 2000 and December 2022, providing a theoretical reference for the investigation of natural products from genus *Cladosporium* and providing a good support for the discovery and development of new drugs.

## 2. Polyketides

### 2.1. Lactones

Four lactone derivatives, 7-hydroxy-3-(2,3-dihydroxybutyl)-1(3*H*)-isobenzofuranone (**1**), isomeric 1-(1,3-dihydro-4-hydroxy-1-isobenzofuranyl) butan-2,3-diols (**2**), 7-hydroxy-1(3*H*)-isobenzo furanone (**3**) and isoochracinic acid (**4**) (Figure 1), were isolated from the solid ferment of the fungus *Cladosporium* NRRL 29097. None of the compounds showed any antifungal activity [32]. (*R*)-Mevalonolactone (**5**) (Figure 1) was isolated from the endophytic fungus *Cladosporium* sp. N5 derived from red alga *Porphyra yezoensis*. No cytotoxicity was found in the brine shrimp lethality test. The result indicated that the crude extract of *Cladosporium* sp. N5 has no toxicity to the aquatic ecosystem. Thus, *Cladosporium* sp. could be used as a potential biocontrol agent to protect the alga from pathogens [33].

(*S*)-5-((1*S*,7*S*)-1,7-dihydroxyoctyl)furan-2(5*H*)-one (**6**) (Figure 1) was isolated from the *Cladosporium* sp. isolated from the Red Sea sponge *Niphates rowi* [34]. (*S*)-5-((1*R*,7*R*)-1,7-dihydroxyoctyl)furan-2(5*H*)-one (**7**) (Figure 1) was isolated from a gorgonian-derived *Cladosporium* RA07-1 collected from the South China Sea [35]. Cladospolide F (**8**),11-hydroxy-γ-dodecalactone (**9**) and (*S*)-5-((1*S*,7*R*)-1,7-dihydroxyoctyl) furan-2(5*H*)-one (**10**) (Figure 1), were isolated from a soft coral-derived fungus *Cladosporium* sp. TZP-29. Compound **8** exhibited moderate inhibitory activity against human pathogenic bacterium *Staphylococcus aureus* (MRSA) with a MIC value of 8.0 μg/mL and acetylcholinesterase inhibitory activity with an IC_50_ value of 40.26 μM [36]. Compound **9** showed potent lipid-lowering activity in HepG2 hepatocytes with an IC_50_ value of 8.3 μM indicating a promising antihyperlipidemic application [25].

Two new cytotoxic 12-membered macrolides, sporiolide A (**11**) and sporiolide B (**12**) (Figure 1), were isolated from the cultured broth of *Cladosporium* sp. L037, which was separated from an Okinawan marine brown alga *Actinotrichia fragilis*. Compounds **11** and **12** exhibited cytotoxicity against murine lymphoma L1210 cells with IC_50_ values of 0.13 and 0.81 μg/mL, respectively. Compound **11** showed antifungal activity against *Candida albicans*, *Cryptococcus neoformans*, *Aspergillus niger* and *Neurospora crassa* with IC_50_ values of 16.7, 8.4, 16.7 and 8.4 μg/mL, respectively, and compounds **11** and **12** exhibited antibacterial activities against *Micrococcus luteus* with IC_50_ values of 16.7 and 16.7 μg/mL, respectively [37]. Additionally, Gesner et al. [34] found one new hexaketide lactone pandangolide 1a (**13**) and its diastereomer pandangolide 1 (**14**) (Figure 1) from the ethyl acetate extract of a *Cladosporium* sp. that was isolated from the Red Sea sponge *Niphates rowi*. Additionally, cladospolide D (**15**), cladospolide A (**16**) and cladospolide B (**17**) (Figure 1), were isolated from the endophytic fungus *Cladosporium* sp. FT-0012. Cladospolide D is a new 12-membered macrolide antibiotic that exhibited excellent antifungal activities against *Pyricularia oryzae* and *Muco rracemosus* with IC_50_ values of 0.15 and 29 μg/mL, respectively [22]. On the basis of biosynthetic considerations (Figure 2), a plausible suggestion for compounds **13** and **14** is that they share a common trihydroxydodecanoic acid-polyketide precursor. The absolute structures of C-11 in **13** and **14** are suggested to be *S*, similar to that of iso-cladospolide B (**6**), considering the function of type I modular PKS in generating a polyketide chain [34].

Two new 12-membered macrolides (**18** and **19**) (Figure 1) were isolated from the plant endophytic fungus *Cladosporium* sp. IFB3lp-2, neither of which exhibited antitumor, antiviral, or acetylcholinesterase inhibitory activities [38]. Five 12-membered macrolides, cladospolide B (**20**), dendrodolide A (**21**), dendrodolide C (**22**), dendrodolide M (**23**) and dendrodolide L (**24**) (Figure 1) were isolated from the gorgonian *Anthogorgia ochracea* (GXWZ-07) derived fungus *Cladosporium* sp. RA07-1 collected from the South China Sea [35]. All the isolated compounds **20**–**24** were evaluated for their antibacterial activities against a panel of pathogenic bacteria, including *Bacillus cereus*, *Tetrag enococcus halophilus*, *S. epidermidis*, *S. aureus*, *Escherichia coli*, *Pseudomonas putida*, *Nocardia brasiliensis* and *Vibrio parahaemolyticus*. Compounds **21**–**23** showed antibacterial activities against all of the tested pathogenic bacteria, with MIC values ranging from 3.13 to 25.0 μM. Among the active compounds, **21** and **22** exhibited the most significant antibacterial activities against *T. halophilus* (MIC = 3.13 μM).

Two new polyketide metabolites, the 12-membered macrolide 4-hydroxy-12-methyloxacyclododecane-2,5,6-trione (**25**) and 12-methyloxacyclododecane-2,5,6-trione (**26**) (Figure 1), were isolated from the endophytic fungal *Cladosprium colocasiae* A801 of the plant *Callistemon viminalis*. Neither of which exhibited cytotoxic activity, antibacterial activity or α-glucosidase inhibition [39]. One new macrolide compound named thiocladospolide E (**27**) and a novel macrolide lactam named cladospamide A (**28**), along with cladospolide B (**29**) (Figure 1), were isolated from mangrove endophytic fungus *Cladosporium* sp. SCNU-F0001. None of the three compounds showed antitumor cell proliferation activity or antibacterial activity [40]. Cladospolide B (**30**) (Figure 1) was isolated from endophytic fungus *Cladosporium* sp. IS384. Compound **30** exhibited antibacterial activity against *Enterococcus faecalis* ATCC 29212 with a MIC value of 0.31 μg/mL [41]. Three new macrolides, cladocladosin A (**31**), thiocladospolide F (**32**) and thiocladospolide G (**33**) (Figure 1), were isolated from the marine mangrove-derived endophytic fungus, *Cladosporium cladosporioides* MA-299 [42]. Compounds **31**–**33** displayed activities against the aquatic pathogenic bacteria *Edwardsiella tarda* and *Vibrio anguillarum* with MIC values ranging from 1.0 to 4.0 μg/mL [43]. Moreover, compound **31** demonstrated activity against aquatic pathogenic bacterium *Pseudomonas aeruginosa*, while compound **32** exhibited activity against plant-pathogenic fungus *Helminthosporium maydis*, both with MIC values of 4.0 μg/mL. A plausible biosynthetic pathway for compounds **31**–**33** as well as the related congeners such as thiocladospolides A–D and pandangolide 3 was proposed, as shown in Figure 3. Briefly, compounds **32** and **33**, thiocladospolides A–D and pandangolide 3 might be derived from the possible precursors dehydroxylated-patulolide C (I), patulolide C (II) or patulolide A (III) through a Michael addition, followed by further oxidation or reduction. The difference was the adding position and attack orientation of the nucleophile (sulfide group) upon the types of the substituent groups at C-4. Generally, when there is no substituent group or the substituent group is only a hydroxyl at C-4, the sulfide group prefers to be added at C-3 to generate compounds **32** and **33**, resulting in the sulfide group at C-3 and methyl group at C-11 located on the opposite faces of the molecule. However, when the substituent group at C-4 is a ketone carbonyl at C-4, the adding position of the sulfide group would be at C-2 to yield thiocladospolide A, resulting in the sulfide group at C-2 and methyl group at C-11 located on the same face of the molecule. Adding different sulfide groups such as methyl 2-hydroxy-3-sulfanylpropanoate and methyl 2-sulfanylethanoate to C-2 of the precursor III, would generate thiocladospolides B and C, respectively. Moreover, compound **31** might be derived via oxidation, cyclization and rearrangement from precursor III.

The strategy of diverted total synthesis (DTS) (Figure 4) has gained popularity as a method of obtaining compounds otherwise unavailable from nature or through the manipulation of the parent compounds [44,45]. The 12-membered macrolactone class of compounds is uniquely well-positioned for exploitation via DTS. These compounds typically have modular synthetic routes, where fragments are synthesized convergently, coupled and cyclized [46,47,48,49]. Side-chain decorations can either be carried through as part of a coupling fragment or synthesized concurrently and coupled to the macrocycle at a later stage. The ability to synthesize these molecules in such a way offers several advantages. Firstly, individual reactions can be modulated to alter stereochemistry and substitution, thus providing flexibility in the synthesis process. Secondly, by using different coupling partners to generate the core structure, it becomes possible to create structural libraries of analogs. Finally, late-stage manipulation through oxidation and substitution reactions allows for the alteration of specific functional group moieties.

Brefeldin A (**34**) (Figure 5) was a new macrolide isolated from the liquid fermentation broth of the cork oak endophytic fungus *Cladosporium* sp. I(R)9-2. Compound 34 was tested for antifungal activity against *Aspergillus niger*, *Candida albicans* and *Trichophyton rubrum*, demonstrating significant antifungal activities with MIC values of 0.97 μg/mL, 1.9 μg/mL and 1.9 μg/mL, respectively [50]. Brefeldin A has been touted as a promising lead molecule in the world of drug development because of its potent biological activity in the antitumor [51] and antiviral [52] fields. Two macrolide compounds, 5Z-7-oxozeaenol (**35**) and zeaenol (**36**) (Figure 5), were isolated from the fermentation broth of the fungus *Cladosporium oxysporum* DH14, a fungus residing in the gut of the Chinese rice locust *Oxya chinensis*. Both compounds exhibited potent phytotoxic activities against the radicle growth of *Amaranthus retroflexus* L. with IC_50_ values of 4.80 and 8.16 μg/mL, respectively, which are comparable to those of the positive control 2,4-dichlorophenoxyacetic acid (IC_50_ = 1.95 μg/mL) [53].

The polyketides cladosporin (**37**), isocladosporin (**38**), 5′-hydroxyasperentin (**39**) and cladosporin-8-methyl ether (**40**) (Figure 5) were isolated from the endophytic fungus *Cladosporium cladosporioides* [54]. Compounds **39** and **40** were first discovered in *C. cladosporioides*. Additionally, 5′,6-diacetylcladosporin (**41**) (Figure 5) was synthesized by acetylating compound **39**. These compounds, **37**–**41**, were evaluated for their antifungal activities against plant pathogens. Compound **37** inhibited the growth of *Colletotrichum acutatum*, *Colletotrichum fragariae*, *Colletotrichum gloeosporioides* and *Phomopsis viticola* by 92.7%, 90.1%, 95.4% and 79.9%, respectively, when tested at 30 µM. Compound **38** showed 50.4%, 60.2% and 83.0% growth inhibition against *C. fragariae*, *C. gloeosporioides* and *P. viticola*, respectively, at the same concentration. Liu Huanhuan and colleagues [55] extracted two compounds, 6,8-dihydroxy-3-methyl-1*H*-isochroman-1-one (**42**) and 6-methoxy-8-hydroxy-3-methylisocoumarin (**43**) (Figure 5), from the solid fermentation of *Cladosporium cladosporioides*. Five isocoumarins (**44**–**48**), 6,8-dihydroxy-4-(*I*-hydroxyethyl)-isocoumarin (**44**), sescandelin B (**45**), 6-hydroxy-8-methoxy-3-methylisocoumarin (**46**), 6-hydroxy-8methoxy-3,4-dimethylisocoumarin (**47**) and aspergillumarin A (**48**) (Figure 5), were isolated from the culture extract of *Cladosporium* sp. JS1-2, an endophytic fungus obtained from the mangrove plant *Ceriops tagal* [56]. Among these, compounds **47** and **48** exhibited inhibitory effects on the growth of cotton bollworm larvae *Helicoverpa armigera* Hubner with IC_50_ values of 100 μg/mL.

### 2.2. Quinone

Five cladosporols, cladosporol (**49**), cladosporol B (**50**), cladosporol C (**51**), cladosporol D (**52**) and cladosporol E (**53**) (Figure 6), were isolated from the fermentation broth of *Cladosporium tenuissimum*, a known hyperparasite of several rust fungi [57]. Compounds **49**–**53** were active in inhibiting the urediniospore germination of the bean rust agent *Uromyces appendiculatus*. Compound **50** was the most active, since germination was completely inhibited at 50 ppm, significantly reduced to more than 90% at 25 ppm, and still low at 12.5 ppm. Compound **49** was more active than compounds **51**–**53**, reaching an inhibition value higher than 80% at the highest concentration. These compounds, by contributing to the reduction of rust survival structures (number and longevity of spores), are expected to play major roles in the multiple aspects of *C. tenuissimum* biocontrol.

One new dimeric tetralone cladosporone A (**54**) and three known analogues cladosporones B–D (**55**–**57**) (Figure 6) were isolated from fungus *Cladosporium* sp. KcFL6′ derived from mangrove plant *Kandelia candel* [19]. Cladosporone A (**54**) inhibits colon cancer cell proliferation by modulating the p21^waf1/cip1^ expression [62]. None of the compounds showed antifungal activity. From a biosynthetic aspect, compounds **54**–**56** could be generated from hexaketide or pentaketide to form the key monomer tetralone, and then the tetralone coupled to yield the key intermediate **56** (Figure 7).

Seven quinones, cladosporol (**58**), cladosporols B–D (**59**–**61**), cladosporol F (**62**), one new cladosporol derivative, 2-chloro-cladosporol D (**63**) and one new brominated derivative 2-bromo-cladosporol D (**64**) (Figure 6), were isolated from the fermentation of the plant-associated fungus *Cladosporium* sp. TMPU1621 [58]. The chlorinated derivative **63** did not exhibit anti-MRSA activity, whereas the bromine congener **64** inhibited the growth of MRSA ATCC43300 and MRSA ATCC700698 with the same MIC values of 25 µg/mL, suggesting that the presence of a bromine atom affects antimicrobial activity against MRSA. Compound **59** displayed potent anti-MRSA activities against both strains with MIC values of 3.13 and 12.5 μg/mL, respectively, whereas compounds **58** and **60** showed weaker activities. Five new perylenequinone derivatives, altertoxins VIII–XII (**65**–**69**), as well as one known compound cladosporol I (**70**) (Figure 6), were isolated from the fermentation broth of the marine-derived fungus *Cladosporium* sp. KFD33 from a blood cockle from Haikou Bay, China. Compounds **65**–**70** exhibited quorum sensing inhibitory activities against *Chromobacterium violaceum* CV026 with MIC values of 30, 30, 20, 30, 20 and 30 μg/well, respectively [59].

Altersolanol J (**71**), altersolanol A (**72**) and macrosporin (**73**) (Figure 6) were isolated from the solid-substrate fermentation culture of *Cladosporium* sp. NRRL 29097 [27]. Compounds **72**–**73** inhibited the growth of *B. subtilis*, producing zones of inhibition of 33 and 23 mm, respectively, and compounds **72**–**73** inhibited *S. aureus*, causing inhibition zones of 31 and 20 mm. While the antibacterial activity of **72** and its analogues is well established, no activity appears to have been reported for **73**. Two naphthoquinones, anhydrofusarubin (**74**) and methyl ether of fusarubin (**75**) (Figure 6), were isolated from *Cladosporium* sp. RSBE-3. Compounds **74** and **75** showed potential cytotoxicity against human leukemia cells (K-562) with IC_50_ values of 3.97 μg/mL and 3.58 μg/mL, respectively. Compound **75** (40 μg/disc) showed prominent activities against *S. aureus*, *Escherichia coli*, *Pseudomonas aeruginosa* and *Bacillus megaterium* with an average zone of inhibition of 27 mm, 25 mm, 24 mm and 22 mm, respectively, and the activities were compared with kanamycin (30 μg/disc) [60]. Compounds **74** and **75** might be useful lead compounds for developing potential cytotoxic and antimicrobial drugs. Anthraquinone (**76**) (Figure 6) was isolated from the rice medium culture of mangrove-derived fungus *Cladosporium* sp. HNWSW-1, isolated from the healthy root of *Ceriops tagal* collected at the Dong Zhai Gang Mangrove Reserve in Hainan. Compound **76** displayed inhibitory activity against α-glycosidase with a IC_50_ value of 49.3 ± 10.6 μΜ [61].

### 2.3. Linear Alkane Compounds

One new polyketide, compound **77** (Figure 8), was isolated from the plant endophytic fungus *Cladosporium* sp. IFB3lp-2. Compound **77** showed no cytotoxicity against human colon cancer cell lines SW1116 and HCT116, breast adenocarcinoma cell line MD-MBA-231, lung adenocarcinoma epithelial cell line A549, hepatocellular carcinoma cell line HepG2 and melanoma cell line A375; no antiviral activity against human enterovirus 71 and Coxsachievirus A16 cell lines; and no acetylcholinesterase inhibition, at the concentration of 20 mM [38]. One new compound, cladospolide E (**78**), and two known derivatives, secopatulolides A and C (**79**–**80**) (Figure 8), were isolated from an unidentified soft coral-derived fungus *Cladosporium* sp. TZP-29. Compounds **78**–**80** were non-cytotoxic, and both showed potent lipid-lowering activities in HepG2 hepatocytes with IC_50_ values of 12.1, 8.4 and 13.1 μM [25]. Two new polyketides (**81**–**82**) (Figure 8) were isolated from the mangrove plant *Excoecaria agallocha*-derived fungus *Cladosporium* sp. OUCMDZ-302 [20]. Mannitol (**83**) (Figure 8) was isolated from the endophytic fungi *Cladosporium cladosporioides* [55].

### 2.4. Other Classes of Polyketides

Four new polyketide-derived metabolites, cladoacetal A (**84**), cladoacetal B (**85**), 3-deoxyisoochracinic acid (**86**) and (+)-cyclosordariolone (**87**) (Figure 9), were isolated from the solid-substrate fermentation culture of *Cladosporium* sp. NRRL 29097. Compound **84** inhibited *S. aureus*, displaying inhibition circles of 13 mm. In addition, compound **86** inhibited the growth of *B. subtilis*, producing zones of inhibition of 8 mm [27]. Alternariol (**88**) and alternariol 5-O-methyl ether (**89**) (Figure 9) were isolated from endophytic *Cladosporium* sp. J6 from endangered *Chrysosplenium carnosum* from Tibet [63]. Compounds **88** and **89** were found to inhibit the photosynthetic electron transport chain in isolated spinach chloroplasts at the same concentrations at which its presence reduced the growth constant of a cyanobacterial (*Synechococcus elongatus*, strain PCO6301) model. These compounds may represent a novel lead for the development of new active principles targeting photosynthesis [64].

Lunatoic acid A (**90**) (Figure 9) was isolated from the endophytic fungus *Cladosporium oxysporum* DH14, a locust-associated fungus. Compound **90** exhibited significantly phytotoxic activity against the radicle growth of *Amaranthus retroflexus* L. with an IC_50_ value of 4.51 μg/mL, which is comparable to that of the positive control 2,4-dichlorophenoxyacetic acid (IC_50_ = 1.95 μg/mL). Furthermore, compound **90** showed selective phytotoxic activity with an inhibition rate of less than 22% against the crops of *Brassica rapa* L., *Sorghum durra*, *Brassica campestris* L., *Capsicum annucm* and *Raphanus sativus* L. under a concentration of 100 μg/mL. The synthesis pathways of derivative compounds **90a** and **90b** on the basis of compound **90** are shown in Figure 10. Both derivatives of compound **90** had moderate phytotoxic activity against the radicle growth of *A. retroflexus* L with inhibition rates of 53.17 and 56.14%, respectively, comparable to that of 2,4-D (87.09%), co-assayed as a positive control under a concentration of 100 μg mL^−1^. A comparison of the phytotoxic activities of compounds **90**, **90a** and **90b** may provide useful hints for the understanding of the ability of compound **90** to inhibit the radicle growth of *A. retroflexus* L. These findings suggest that compound **90** has some potential as a new agent for weed control [53].

From the endophytic fungus *Cladosporium* cladosporioides JG-12, five compounds with different structural types were identified [65]. Of these compounds, (5*S*)-5-hydroxy-7-(4″-hydroxy-3″-methoxy-phenyl)-1-phenyl-3-heptanone (**91**) (Figure 9) showed antibacterial activity against *Ralstonia solanacearum* and *S. aureus*. Additionally, this compound exhibited acetylcholinesterase inhibitory activity with an inhibitory rate of 23.54%.

The fungal strains *Cladosporium* sp. NJF4 and NJF6 were collected from marine sediments in the Gulf of Prydz, and their secondary metabolites were isolated and characterized comprehensively. The results of this study yielded a total of 20 compounds, with two of them, 7-deoxy-7,8-didehydrosydonic acid (**92**) and sydonic acid (**93**) (Figure 9), being isolated for the first time from the genus *Cladosporium* [66]. Compound **93** was found to be weakly cytotoxic against HL-60 human promyelocytic leukemia and A-549 human lung carcinoma cell lines. Compound **93** exhibited significant inhibiting activities to four pathogenic bacteria, *Bacillus subtilis*, *Sarcina lutea*, *Escherichia coli* and *Micrococcus tetragenus*, and uniquely against two marine bacterial strains *Vibrio Parahaemolyticus* and *Vibrio anguillarum* [67]. These findings suggest that the marine environment in the Gulf of Prydz may harbor diverse fungal species with the potential to produce unique secondary metabolites. The identification of new compounds from these strains could have significant implications for drug discovery and development.

One new abscisic acid analogue, cladosacid (**94**) (Figure 9), was isolated from the marine-derived fungus, *Cladosporium* sp. OUCMDZ-1635 [68]. Compound **94** did not demonstrate antibacterial activities against *Bacillus subtilis* CGMCC 1.3376, *Clostridium perfringens* CGMCC 1.0876, *Escherichia coli* ATCC 11775, *Pseudomonas aeruginosa* ATCC10145, *S. aureus* ATCC 6538 or *Candida albicans* ATCC 10231, even when tested at concentrations of 100 μg/mL. One prenylated flavanone derivative (**95**) (Figure 9) was isolated from the culture broth of Indonesian marine sponge-derived *Cladosporium* sp. TPU1507 [69]. Compound **95** exhibited moderate inhibitory activity against PTP1B with an IC_50_ value of 11 μM. Therefore, compound **95** shows promise as a potential drug target for treating diseases associated with PTP1B inhibition, such as obesity, diabetes mellitus or cancer.

Three new polyketides (**97**–**99**) and one known compound (**96**) (Figure 9) were obtained from the fermentation products of the endophytic fungus *Cladosporium* sp. OUCMDZ-302, which was derived from the mangrove plant *Excoecaria agallocha* (*Euphorbiaceae*) [20]. Notably, compound **99** exhibited significant radical scavenging activity against DPPH with an IC_50_ value of 2.65 μM, indicating its promising potential as a natural antioxidant agent. Compounds **96** and **97** were postulated to be biosynthesized by the polyketide pathway from acetyl coenzyme A (Figure 11). The pathway involves the condensation, cyclization, dehydration and hydrogenation of acetyl-CoA units. The oxidation and reduction of (*S*)-12 results in the formation of compound **98**. Furthermore, (*S*)-12 underwent Baeyer–Villiger oxidation, followed by methanolysis and hydrolysis, to yield compound **99**. The formations of compounds **75** and **76**, on the other hand, are the results of the condensation, reduction, dehydration and decarboxylation of acetyl-CoA units of different lengths.

Four new citrinin derivatives, cladosporins A–D (**100**–**103**) (Figure 12) were isolated from a culture broth of the deep-sea-derived fungus *Cladosporium* sp. SCSIO z015. Compounds **100**–**103** showed weak toxicity toward brine shrimp naupalii with IC_50_ values of 72.0, 81.7, 49.9 and 81.4 μM, respectively. These values were compared to a positive control, toosendanin, with an IC_50_ value of 21.2 μM. And **103** also showed significant antioxidant activity against DPPH radicals with an IC_50_ value of 16.4 μM. This promising compound holds potential for further development as a natural antioxidant agent [70].

One benzofuran derivative (**105**), one isochroman derivative (**106**) and two other compounds **104** and **107** (Figure 12) were isolated from the culture extract of *Cladosporium* sp. JS1-2, an endophytic fungus obtained from the mangrove plant *Ceriops tagal*. Compounds **105**–**107** displayed antibacterial activities against *S. aureus* with the same MIC values of 12.5 μg/mL; the positive control was ciprofloxacin, with a MIC value of 3.12 μg/mL. Compound **107** displayed growth inhibition activity against newly hatched larvae of *Helicoverpa armigera* Hubner, with the same IC_50_ values of 100 μg/mL; the positive control was azadirachtin, with an IC_50_ value of 50 μg/mL [56].

7,4′-dihydroxyisoflavone (**108**) (Figure 12) was isolated from the Antarctic fungus *Cladosporium* sp. NJF6 [66]. Compound **108** displayed inhibitory activities on inflammatory markers like NF-kB, TGF-β, TNF-α, IL-6, IL-8 and COX-2. It also had an effect on various apoptotic markers like caspase-3, caspase-8, Bcl-2 and Bax, which indicates that compound **108** may have significant effects on breast cancer, cardiovascular diseases, neurodegenerative diseases, diabetes and its complications, osteoporosis, eczema and skin inflammation [72].

Two novel xanthone-derived metabolites, cladoxanthones A (**109**) and B (**110**), along with one known mangrovamide J (**111**) (Figure 12) were discovered in *Cladosporium* sp. QH07-10-13 [71]. The new metabolites **109** and **110** exhibited an intriguing spiro [cyclopentane-1,2′-[3,9a] ethanoxanthene]-2,4′,9′,11′(4a′*H*)-tetraone framework. Compound **109** could be produced from α-methylene ketone and dihydro-xanthone, through a Diels–Alder reaction. Compound **110** could result from the oxidative coupling products of compounds **109** and **111** (Figure 13). Compounds **109**–**111** demonstrated low cytotoxic activities.

There are a total of 111 polyketides that have been isolated from *Cladosporium*. These polyketides are classified as lactones (43.24%), quinones (25.23%), linear alkanes (6.31%) and others (25.23%). Out of the total isolated polyketides, 62 originated from marine sources, while 39 compounds are newly identified compounds. Hence, marine sources make up 66.7% of the newly discovered substances. Additionally, there are 66 bioactive compounds in total, with 36 of them coming from the ocean, accounting for 54.5% of the bioactive substances. Polyketides are significant secondary metabolites of *Cladosporium*, with some natural products exhibiting biological activities concentrating on cytotoxicity, antibacterial activity and antifungal activity.

## 3. Alkaloids

*N*-*β*-acetyltryptamine (**112**) (Figure 14), was isolated from the endophytic fungus *Cladosporium* sp. N5 associated with red alga *Porphyra yezoensis*. No cytotoxicity was found in the brine shrimp lethality test, which indicated that the crude extract of *Cladosporium* sp. has no toxicity to the aquatic ecosystem. Thus, *Cladosporium* sp. can be applied as a biocontrol agent [33]. Six new indole alkaloids including five new glyantrypine derivatives, 3-hydroxyglyantrypine (**113**), 14*R*-2-oxoglyantrypine (**114**), 14*S*-2-oxoglyantrypine (**115**), cladoquinazoline (**116**) and *epi*-cladoquinazoline (**117**), and one new pyrazinoquinazoline derivative, norquinadoline A (**118**), together with eight known alkaloids, quinadoline A (**119**), deoxynortryptoquivaline (**120**), deoxytryptoquivaline (**121**), tryptoquivaline (**122**), CS-C (**123**), quinadoline B (**124**), prelapatin B (**125**) and glyantrypine (**126**) (Figure 14), were isolated from the culture of the mangrove-derived fungus *Cladosporium* sp. PJX-41. Anti-H1N1 activities were measured for these compounds, with compounds **115**, **118**, **120**–**122** and **124** showing noteworthy antiviral activities with IC_50_ values of 85, 82, 87, 85, 89 and 82 μM, comparable to that of the positive control ribavirin (IC_50_ = 87 μM). However, the other alkaloids, **113**, **114**, **116**, **117**, **119**, **123** and **126**, showed weakly antiviral activities (IC_50_ values ranging between 100 and 150 μM), and compound **125** displayed no activity (IC_50_ > 200 μM) [26].

Ma Yanhong’s group [63] isolated *β*-carboline (**127**) and uracil nucleoside (**128**) (Figure 14) from the fermentation of endophytic *Cladosporium* sp. J6 from endangered *Chrysosplenium carnosum* from Tibet. Compound **127** displayed a wide range of biological activities including antitumor [73], antiviral [74] and antimicrobial activity [75]. Three sulfur-containing compounds, cladosporin A (**129**), cladosporin B (**130**) and haematocin (**131**) (Figure 14), were isolated from the marine fungus *Cladosporium* sp. The compounds belong to the class of cyclic di-acid alkaloids, with **129** and **130** being new members. Compounds **129**–**131** displayed moderate cytotoxic activities against the HepG2 cell line, with IC_50_ values of 21, 42 and 48 μg/mL, respectively [76]. Uracil (**132**) (Figure 14) was isolated from the fermentation of endophytic *Cladosporium* sp. J6 from endangered *Chrysosplenium carnosum* from Tibet [63]. Cladosporilactam A (**133**) (Figure 14), one new bicyclic lactam, was isolated from the fungus *Cladosporium* sp. RA07-1, which originated from the gorgonian *Anthogorgia ochracea* (GXWZ-07) derived from the South China Sea [35]. Compound **133** was the first example of a 7-oxabicyclic [6.3.0] lactam obtained from a natural source. Research indicates that compound **133** exhibits potent cytotoxicity against a series of cancer cell lines, including cervical cancer HeLa, mouse lymphocytic leukemia P388, human colon adenocarcinoma HT-29 and human lung carcinoma A549 with IC_50_ values of 0.76, 1.35, 2.48 and 3.11 μM, respectively, which suggests that it might have potential to be developed as an antitumor agent. Three alkaloids, ilicicolin H (**134**), (7*R*)-methoxypurpuride (**135**) and (5a*S*,9*S*,9a*S*)-1,3,4,5,5a,6,7,8,9,9,9a-decahydro-6,6,9a-trimethyl-3-oxonaphtho [1,2-c] furan-9-yl *N*-acetyl-*L*-valinate (**136**) (Figure 14), were isolated from the endophytic fungus *Cladosporium cladosporioides* JG-12. Compounds **135** and **136** showed inhibitory activities against *Panagrellus redivivus*, and compounds **134**–**136** exhibited acetylcholinesterase inhibitory activities [65]. 2′-deoxythymidine (**137**) and 3-carboxylic acid (**138**) (Figure 14) were isolated from the extracts of the culture of sponge *Callyspongia* sp. derived fungus *Cladosporium* sp. SCSIO41007 [28].

**Figure 14 ijms-25-01652-f014:**
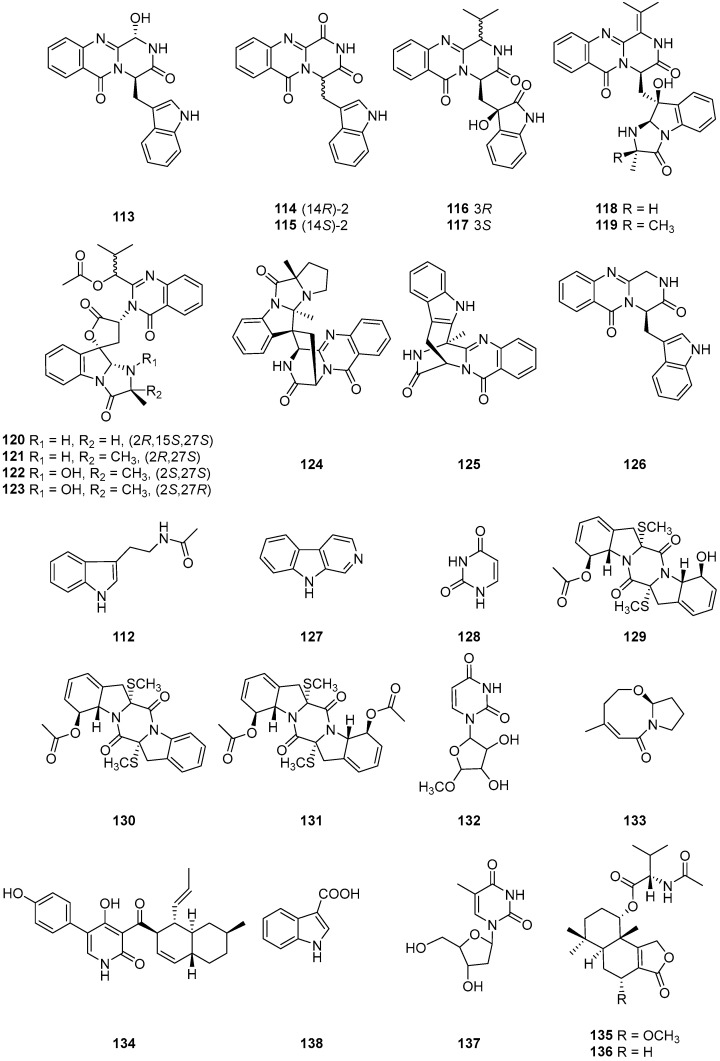
Structures of compounds **112**–**138** [26,28,33,35,63,65,76].

One new alkaloid with a 3-(2*H*-pyran-2-ylidene) pyrrolidine-2,4-dione nucleus, cladodionen (**139**) (Figure 15), was isolated from the marine-derived fungus *Cladosporium* sp. OUCMDZ-1635. Cladodionen (**139**) showed cytotoxic activities against MCF-7, HeLa, HCT-116 and HL-60 human cancer cell lines with IC_50_ values of 18.7, 19.1, 17.9 and 9.1 μM [68]. Cladosporamide A (**140**) (Figure 15), one new protein tyrosine phosphatase (PTP) 1B inhibitor, was isolated from the culture broth of an unidentified Indonesian marine sponge-derived *Cladosporium* sp. TPU1507. Compound **140** modestly inhibited PTP1B and T-cell PTP (TCPTP) activities with IC_50_ values of 48 and 54 μM, respectively [69]. One alkaloid adenine nucleoside (**141**) (Figure 15) was isolated from the solid fermentation of *Cladosporium cladosporioides* [55]. Two new succinimide-containing derivatives, cladosporitin A (**142**) and B (**143**), along with the previously reported talaroconvolutin A (**144**) (Figure 15), were isolated from the fermentation culture of the tree (*Ceriops tagal*) root-derived fungus *Cladosporium* sp. HNWSW-1. Compound **143** exhibited cytotoxicity against BEL-7042, K562 and SGC-7901 cell lines, with IC_50_ values of 29.4 ± 0.35, 25.6 ± 0.47 and 41.7 ± 0.71 μM, respectively. Compound **144** showed cytotoxicity against Hela and BEL-7042 cell lines with IC_50_ values of 14.9 ± 0.21 and 26.7 ± 1.1 μM, respectively. Moreover, compound **144** was found to display enzyme inhibitory activity against *α*-glycosidase, with an IC_50_ value of 78.2 ± 2.1 μM [61]. 3-indoleacetic acid (**145**) and 3-formylindole (**146**) (Figure 15) were isolated from the fungal strains *Cladosporium* sp. NJF6 derived from marine sediments in the Gulf of Priz, Antarctica [66]. Compound **146** caused inhibition on the growth of *Trypanosoma cruzi*, with an IC_50_ value of 26.9 μM, with moderate cytotoxicity against Vero cells. And compound **146** was found to be inactive when tested against *Plasmodium falciparum* and *Leishmania donovani*, therefore showing selectivity against *T. cruzi* parasites [77]. Eight new tetramic acid derivatives, cladosporiumins A–H (**147**–**154**) (Figure 15), were isolated from a culture broth of *Cladosporium* sp. SCSIO z0025 derived from deep-sea sediment collected from Okinawa Trough. Despite their novelty, assays for antitumor cytotoxic, antibacterial and antiacetylcholinesterase activities demonstrated that these compounds exhibited negligible inhibitory effects [78]. One indole diterpenoid alkaloid, cladosporine A (**155**) (Figure 15), has been identified and isolated from the fungus *Cladosporium* sp. JNU17DTH12-9-01 [79], marking the first discovery of such an alkaloid within the genus *Cladosporium*. This new alkaloid has demonstrated antimicrobial activities against *S. aureus* 209P with a MIC of 4 μg/mL, and a MIC of 16 μg/mL against *Candida albicans* FIM709.

A total of 44 alkaloids have been identified from the genus *Cladosporium*. The structural types of these alkaloids include amines, indoles, pyrrolizidines and quinazolines. Out of the total isolated alkaloids, 36 originated from marine sources. And 21 compounds are newly identified compounds, with marine sources accounting for 66.7% of these newly discovered substances. Furthermore, all of this new material comes from the ocean. In total, there are 27 bioactive compounds, with 22 of them derived from marine sources, making up 81.5% of the bioactive substances. These alkaloids exhibit biological activities, focusing on antiviral, cytotoxicity and enzyme inhibition activities.

## 4. Steroids and Terpenoids

One new pentanorlanostane derivative, cladosporide A (**156**), along with 23,24,25,26,27-pentanorlanost-8-ene-3*β*,22-diol (**157**), 18,22-cyclosterols (**158**), tetranorditerpenoids (**159**), ketodialdehyde derivative (**160**), ketoaldehyde carboxylic acid (**161**), lanosterol (**162**), hexanorlanosterols (**163**) and (**164**), pentanorlanostanecarboxylic acid derivatives, 3*β*-hy-droxy-4,4,14*α*-trimethyl-5*α*-pregna-7,9(11)-diene-20*S*-car-boxylic acid (**165**), 3*β*,12*β*-dihydroxy-4,4,14*α*-trimethyl-5*α*-pregna-7,9(11)-diene-20*S*-carboxylic acid (**166**) and 4,4,14*α*-trimethyl-3-oxo-5*α*-pregna-7,9(11)-diene-20*S*-car-boxylic acid (**167**) (Figure 16) were isolated from *Cladosporium* sp. IFM 49189 [13]. Cladosporide A (**156**) showed the characteristic inhibition against a human pathogenic filamentous fungus, *Aspergillus. fumigatus*, at 6.25 μg/disc, whereas no inhibition was observed against a pathogenic filamentous fungus, *A. niger*, or pathogenic yeasts, *C. albicans* and *C. neoformans*. Compound **156** did not completely inhibit the growth of *A. fumigatus*, but **156** apparently reduced the growth of this fungus. Thus, compound **156** has been identified as a characteristic antifungal agent against the human pathogenic filamentous fungus *A. fumigatus*. Hosoe et al. [80] extended their research to isolate an additional set of several triterpenoids, including cladosporide A (**156**), new pentanorlanostane derivatives, cladosporides B–D (**168**–**170**), 23,24,25,26,27-pentanorlanost-8-ene-3*β*,22-diol (**171**), dihydrocladosporide A (**172**) and 3,30-dioxo-23,24,25,26,27-pentanorlanost-8-en-22-0ic acid (**173**) (Figure 16) from *Cladosporium* sp. IFM 49189. Compounds **156** and **168** strongly inhibited the growth of *A. fumigatus* (1.5 and 3.0 ug/disc, respectively), whereas compounds **169** and **170** showed no antifungal activity. Compounds **172** and **173** retained weak inhibitory activity against *A. fumigatus*. Zou et al. [81] identified three steroids, ergosta-5,7,22-triene-3*β*-ol (**174**), eburicol (**175**) and *β*-sitosterol (**176**) (Figure 16), from *Cladosporium cladosporioides*, an endophytic fungus derived from an unidentified mangrove. One steroidal compound, ergosterol peroxide (3*β*-hydroxy-5,8-epidioxy-ergosta-6,22-diene) (**177**) (Figure 16), was isolated from the endophytic fungus *Cladosporium cladosporioides* JG-12 derived from *Ceriops tagal* [65]. Compound **177** exhibited potent anti-inflammatory activity [82]. It has been reported that ergosterol peroxide has the potential for inhibiting cancer cell proliferation as well as inducing apoptosis in cancer cells [83,84,85].

Three new highly oxygenated sterols, cladosporisteroids A–C (**178**–**180**) (Figure 17), together with three known compounds, pregn-7-dien-3,6,20-trione (**181**), 3*β*,5*α*,9*α*-trihydroxy-(22*E*,24*R*) ergosta-7,22-diene-6-one (**182**) and cerevisterol (**183**) (Figure 17), were isolated from the extracts of the culture of the sponge *Callyspongia* sp. derived fungus *Cladosporium* sp. SCSIO41007 [28]. Antiviral and cytotoxic activities of compounds **178**–**183** were tested against H3N2 and EV71 viruses, as well as the K562, MCF-7 and SGC7901 cancer cell lines. None of the tested compounds showed any cytotoxic effects on the cancer cell lines. However, compound **178** exhibited weak inhibitory activity against H3N2 with an IC_50_ value of 16.2 μM, while the IC_50_ value for the positive control oseltamivir was 34.0 μM.

*Cladosporium* sp. WZ-2008-0042, a fungus obtained from a gorgonian *Dichotella gemmacea* collected from the South China Sea, produced one new pregnane, 3*α*-hydroxy-7-ene-6,20-dione (**184**), along with five known steroids, including 5*α*,8*α*-epidioxy-ergosta-6,9,22*E*-triene3*β*-ol (**185**), 5*α*,8*α*-epidioxy-ergosta-6,22*E*-dien-3*β*-ol (**186**), ergosta-7,22*E*-diene-3*β*,5*α*,6*β*-triol (**187**), 3*β*,5*α*-dihydroxy-6*β*-methoxyergosta7,22-diene (**188**) and ergosterol (**189**), and one known steroidal glycoside (**190**) (Figure 17) [21]. Compounds **184**–**186** and **188** exhibited antiviral activities against respiratory syncytial virus (RSV). Specifically, compound **184** exhibited potential antiviral activity against RSV, with an IC_50_ value of 0.12 mM. Additionally, compound **189** demonstrated moderate antibacterial activity against *Shigella dysenteriae*, with a MIC value of 3.13 μM. This discovery presents a promising avenue for further research on the antiviral and antibacterial properties of these compounds, as well as their potential therapeutic applications. Ma Chuan et al. [41] obtained ergosterol (**191**) (Figure 17) from the fermentation broth of an endophytic fungus *Cladosporium* sp. IS384. Compound **191** showed weak cytotoxic and antioxidant activities [86]. The dose effect relationship of ergosterol to nerve cell SH-SY5Y neurotoxicity was not significant at 6.25–25 μg/mL, but it had better effect on the oxidative damage protection of nerve cell SH-SY5Y at 6.25 μg/mL. And ergosterol can inhibit the activity of apoptosis protein caspase 3 to achieve the antioxidative protection effect of nerve cell SHSY5Y [87]. Additionally, two steroids, myristate-4-en-3-one (**192**) and 3*β*-hydroxy-5*α*,8*α*-peroxidized ergot-6,22-diene (**193**) (Figure 17), were isolated from the endophytic fungi *Cladosporium cladosporioides* [55]. The fungus *Cladosporium* sp. NJF4 was found to produce 5,22-diene-ergosta-3*β*,7*β*,8*β*-triol (**194**) (Figure 17) [66]. Compound **194** was found to induce cytotoxicity with a MIC value of 14.1 μM in HeLa cells in vitro [88].

A total of 39 steroids and terpenoids have been reported in the genus *Cladosporium*, with steroids being more common and having keratosteroid skeletons, while terpenoids are more commonly tetracyclic triterpenoids with lanolin steroid skeletons. Out of the total isolated steroids and terpenoids, 17 are derived from marine sources. Additionally, 11 compounds are newly identified, with marine sources accounting for 36.4% of these newly discovered substances. These steroids and terpenoids exhibit various biological activities, including antifungal, antiviral, cytotoxicity, and enzyme inhibition activities. In total, there are 10 bioactive compounds, with marine sources contributing to 40% of these bioactive substances.

## 5. Benzene Derivatives

One new polyketide-derived metabolite, 3-(2-formyl-3-hydroxyphenyl)-propionic acid (**195**) (Figure 18), was isolated from solid-substrate fermentation cultures of *Cladosporium* sp. NRRL 29097. Compound **195** showed growth inhibition activity against *Bacillus subtilis* with an inhibition circle of 22 mm [27]. Seven benzene derivatives, *L*-*β*-phenyllactic acid (**196**), *α*-resorcylic acid (**197**), *p*-hydroxy benzoic acid methyl ester (**198**), phenyllactic acid (**199**), 4-hydroxyphenyl alcohol (**200**), *p*-hydroxyphenylacetic acid (**201**) and *p*-hydroxybenzyl alcohol (**202**) (Figure 18), were derived from the endophytic fungus *Cladosporium* sp. N5 associated with red alga *Porphyra yezoensis*. It is significant to note that none of the identified compounds show any toxic effects on brine shrimps, which indicates that the environment-friendly *Cladosporium* sp. could be used as a potential biocontrol agent to protect the alga from pathogens [33]. Compound **199** was reported to be active against various bacteria and fungi [89,90], and compound **196** was reported to be a strong fungicide [91]. They may play important roles in developing the symbiotic relationship between plants and microbes. Compounds **199** and **201** were reported to inhibit the growth of red alga *Porphyra tenera* conchocelis [92]. Compound **203** (Figure 18), 2-chloro-3,5-dimethoxybenzyl alcohol, was isolated from the endophytic fungus *Cladosporium cladosporioides* JG-12. It displayed significant inhibitory effects against *Candida albicans*, and also showed inhibitory activities against *Panagrellus redivivus* and acetylcholinesterase. This finding suggests that compound **203** may have potential therapeutic applications for the treatment of fungal infections, such as *Staphylococcus aureus*, *Canidia albicans*, *Ralstonia solanacearum* and nematode *Panagrellus redivivus* infestations, and neurodegenerative diseases [65]. Citrinin H2 (**204**) and *N*-(4-hydroxy-2-methoxyphenyl) acetamide (**205**) (Figure 18) were isolated from the culture extract of *Cladosporium* sp. JS1-2, an endophytic fungus obtained from the mangrove plant *Ceriops tagal*. Compound **204** displayed antibacterial activity against *S. aureus* with the same MIC values of 12.5 μg/mL. Compounds **204** and **205** displayed growth inhibition activities against newly hatched larvae of *Helicoverpa armigera* Hubner, with the same IC_50_ values of 100 μg/mL, and the positive control was azadirachtin, with a IC_50_ value of 50 μg/mL [56]. The Antarctic fungus *Cladosporium* sp. derived NJF6 and NJF4 were found to produce six benzene derivative analogues, *N*-acetyl phenethylamine (**206**), phenethylamine (**207**), and *p*-hydroxyphenethylamine (**208**), *p*-hydroxyphenyl propionic acid (**209**), 1,2-benzenedicarboxylic acid (**210**) and benzoic acid (**211**) (Figure 18) [66].

A total of 17 benzene derivatives were identified from the ferments of *Cladosporium* sp., with 15 of these compounds originating from marine sources. Among the total isolated benzene derivatives, there are a total of eight bioactive compounds, with seven of them derived from oceanic sources. Notably, the oceanic compounds account for 87.5% of the bioactive substances. These benzene derivatives exhibit biological activities such as antibacterial, phytotoxicity and insecticidal activities.

## 6. Cyclic Peptides

Cyclo (Trp-Pro) (**212**) and cyclo (Trp-Val) (**213**) (Figure 19) were derived from the endophytic fungus *Cladosporium* sp. N5 associated with red alga *Porphyra yezoensis* [33]. Cyclo (Gly-Leu) (**214**) (Figure 19) was isolated from the extracts of the culture of the sponge *Callyspongia* sp. derived fungus *Cladosporium* sp. SCSIO41007 [28]. An antimicrobial study indicated that compound **214** has therapeutic potential as an antibacterial and antifungal agent. In an anticancer study, cyclo (Gly-Leu) exhibited moderate activities in inhibiting various cancer cell lines including HT-29, MCF-7 and HeLa cells [93]. Cyclic (phenylalanine-aspartic acid) (**215**), cyclic (tryptophan-aspartic acid) (**216**), cyclic (tryptophan-aspartic acid) (**217**), 4-hydroxyphenylalanine-leucine (**218**), cyclic (proline-tyrosine) (**219**), cyclic (proline-tyrosine) (**220**) and cyclic (valine-proline) (**221**) (Figure 19) were isolated from the fungal *Cladosporium* sp. NJF4 and NJF6, derived from marine sediments in the Gulf of Priz, Antarctica [66].

Ten cyclic peptides were identified from the ferments of *Cladosporium* sp. These compounds originate from the ocean. However, only compound **214** demonstrates growth inhibitory activity, while the biological activities of the remaining compounds require further investigation.

## 7. Others

Nicotinic acid (**222**) and acetyl-tyramine (**223**) (Figure 20) were obtained from the endophytic fungus *Cladosporium* sp. N5 associated with red alga *Porphyra yezoensis* [33]. 1,8-dimethoxynaphthalene (**224**) (Figure 20) was isolated from the culture extract of *Cladosporium* sp. JS1-2, an endophytic fungus obtained from the mangrove plant *Ceriops tagal*. Compound **224** displayed antibacterial activity against *S. aureus* with a MIC value of 12.5 μg/mL; the positive control was ciprofloxacin, with a MIC value of 3.12 μg/mL. It also displayed growth inhibition activity with an IC_50_ value of 100 μg/mL against newly hatched larvae of *Helicoverpa armigera* Hubner; the positive control was azadirachtin, with an IC_50_ value of 50 μg/mL [56,94]. Four new cyclohexene derivatives, cladoscyclitols A–D (**225**–**228**), and one new ribofuranose phenol derivative, 4-*O*-*α*-*D*-ribofuranose-2-pentyl-3-phemethylol (**229**) (Figure 20), were obtained from the mangrove-derived endophytic fungus *Cladosporium* sp. JJM22. Compounds **226** and **229** displayed potent inhibitory activities against *α*-glucosidase with IC_50_ values of 2.95 and 2.05 μM, respectively [23].

Eight bioactive compounds with antibacterial, insecticidal, and enzyme inhibitory activities were discovered in the ferments of *Cladosporium* sp., all originating from the ocean. This further highlights the significance of the ocean as an invaluable resource.

## 8. Conclusions

From January 2000 to December 2022, a total of 229 natural products were isolated from the genus *Cladosporium* (Table 1), of which 64.63% of the compounds were isolated from the ocean and 34% of the compounds were found for the first time. These findings strongly suggest that marine-derived *Cladosporium* has great potential to produce abundant compounds with new structures. Before 2007, there were few studies on the secondary metabolites produced by *Cladosporium*, and their structural types were mainly concentrated on terpenoids and polyketides. After 2008, studies on the natural products isolated from *Cladosporium* gradually increased, with the number of compounds isolated each year on overall upward trend, and the structural types of the isolated compounds were gradually diversified, including cyclic peptides, alkaloids and some other types of compounds, indicating that the genus *Cladosporium* has the potential to produce compounds of multiple structural types (Figure 21). The structures of the isolated compounds with diverse skeletons are mainly concentrated in the classes of polyketides, alkaloids, steroids, terpenoids, benzene derivatives and cyclic peptides (Figure 22). Polyketides, which make up 48% of the natural products derived from this genus, are notably significant among these compounds.

The sources of *Cladosporium* are distributed in different ecosystems, including the Antarctic, forests and oceans. About 65% of the isolated natural products were separated from marine organism-derived *Cladosporium*, including sponge, mangrove and gorgonian. The first marine-derived natural product was isolated from an unidentified sponge-derived *Cladosporium* in 2001 [22]. A larger number of compounds, excluding steroids and terpenoids, have been isolated from *Cladosporium* in the ocean compared to those obtained from land (Figure 22), which indicates that marine fungi *Cladosporium* have great potential to produce compounds.

The genus *Cladosporium* has the potential to produce a great diversity of bioactive secondary metabolites, including antibacterial, cytotoxic, growth inhibitory, enzyme inhibitory, antifungal activity, and quorum sensing inhibitory activity (Figure 22, Table 2, Table 3 and Table 4). The bioactive compounds isolated from the genus *Cladosporium* mainly focus on antibacterial activity (26%), cytotoxicity (16%) and antiviral activity (12%), indicating considerable potential for the development of new antibiotics and anticancer compounds from *Cladosporium*. In addition, many compounds with diverse bioactivities, especially cytotoxic, antibacterial, antiviral and quorum sensing inhibitory compounds are predominantly found in the ocean (Figure 22). These findings emphasize the ocean as a valuable resource and propose that the marine genus *Cladosporium* has the capacity to generate numerous secondary metabolites with various bioactivities.

The genus *Cladosporium* is capable of producing various secondary metabolites with diverse bioactivities, including antibacterial activity, cytotoxicity, antifungal activity, enzyme inhibition activity, antiviral activity, quorum sensing inhibitory activity and antioxidant activity (Figure 22). Research shows that 63% of the natural products derived from *Cladosporium* exhibit bioactive activities. Among these compounds, **11**, **34**, **127**, **191**, **199** and **203** have demonstrated more than three types of activities (Figure 22, Table 2, Table 3 and Table 4), highlighting the potential of this genus to produce bioactive natural products. Additionally, it is noteworthy that 59% of these active compounds are isolated from marine-derived fungi, further supporting the development prospects of marine fungi.

Structure activity relationships (SARs) can be used to predict biological activity from molecular structure. Wang et al. [54] reported an evaluation of the relationships between structure and bioactivity for cladosporin (**37**) and its analogues (**38**–**41**). After an overall evaluation of the relationship between the structures and antifungal activity of the compounds at 30 μM, several essential positions were identified as potential determinants of their antifungal activity. The absolute configuration of C-6′ in the structures of compounds **37** and **38** was found to have a significant impact on the antifungal activity of the parent compound. Specifically, the *R* configuration of C-6′ in structure **38** led to a marked decrease in antifungal activity against *Colletotrichum* species, while slightly increasing the antifungal activity against *Phomopsis* species. Comparing the structures of compounds **37** and **39** revealed that he introduction of a hydroxyl group at the C-5′ position results in a complete loss of antifungal activity against *Colletotrichum* species and decreased selectivity against *Phomopsis* species, highlighting the importance of maintaining an unsubstituted C-5′ for antifungal activity. Furthermore, by comparing the structures of compounds **37** and **40**, it was observed that the replacement of the hydroxyl group with a methoxy group at C-8 caused a broad loss of antifungal activity against all tested fungi, suggesting that this position might be the active site where hydrogen bonds are formed. Additionally, when compounds **39** and **41** were compared, the replacement of the hydrogen of the hydroxyl group at C-6 and the hydrogen at C-5′ with acetyl groups greatly increased the selectivity toward the two *Phomopsis* species. Therefore, the differences in activity indicated that the *S* configuration of C-6′, the openness of C-5′, the presence of a hydroxyl group at C-8 and the introduction of functional groups at C-6 influence the antifungal properties of these compounds [54].

Compounds **9**, **78**, **79** and **80** exhibit potent lipid-lowering activities in HepG2 hepatocytes (Figure 23, Table 4), suggesting that they can be developed into hypoglycemic agents (Figure 23). Compound **34** displays significant antifungal activity (Figure 23, Table 4), declaring the potential of **34** to be applied in agricultural fungicide. Compounds **35**, **36** and **90** demonstrate potent phytotoxic activities against the radicle growth of *Amaranthus retroflexus L* (Figure 23, Table 4). This indicates the potential for developing compounds **35**, **36** and **90** as new herbicides. Compound **59** demonstrates a stronger antibacterial activity against MRSA than the positive control (Figure 23, Table 2), highlighting the challenge of bacterial drug resistance. Compounds **75**, **93**, **156** and **168** display potent antibacterial activities compared to the positive control (Figure 23, Table 2), which means they could be valuable starting points for the development of new antibiotics. Compounds **115**, **118**, **120**–**122** and **124** show noteworthy antiviral activities (Figure 23, Table 4), which support their potential use as antibiotics. Compound **133** exhibits potent cytotoxicity against a series of cancer cell lines (Figure 23, Table 4), including cervical cancer HeLa, mouse lymphocytic leukemia P388, human colon adenocarcinoma HT-29 and human lung carcinoma A549 with IC_50_ values of 0.76, 1.35, 2.48 and 3.11 μM, respectively, which suggests that it might have potential to be developed as an antitumor agent. Compound **177** exhibits potent anti-inflammatory activity (Figure 23, Table 4), declaring the potential of **177** to be applied in adjuvant drugs for anticancer therapy. Compound **184** displays potential antiviral activity against RSV (Figure 23, Table 4), which demonstrates that **184** could be employed in developing vaccines and antiviral drugs. Compound **196** exhibits noteworthy antibacterial activity (Figure 23, Table 2), making it a promising candidate for developing a strong fungicide. Compound **203** displays significant inhibitory effects against *Candida albicans*, and also showed inhibitory activities against *Panagrellus redivivus* and acetylcholinesterase (Figure 23, Table 2, Table 3 and Table 4). This finding suggests that compound **203** may have potential therapeutic applications for the treatment of fungal infections, nematode infestations and neurodegenerative diseases. Compounds **226** and **229** exhibit strong inhibition against *α*-glucosidase (Figure 23, Table 4), indicating their potential use in antidiabetic therapy. These results further suggest that the genus *Cladosporium* holds promise as a source of bioactive compounds.

The secondary metabolites of *Cladosporium* may play crucial roles in the ecosystem and have specific ecological effects. Some secondary metabolites, such as compounds **115**, **120**–**122**, **124** and **156**, have antifungal and antiviral effects, which can be utilized for biological control, managing the growth and reproduction of pests and pathogenic microorganisms and safeguarding crops and forest vegetation [13,21,26,44]. Some compounds, like compounds **35**, **36** and **90**, exhibit potent phytotoxicity and show promise as new herbicides (Figure 23) [53]. Volatile organic compounds can influence plant communication, aid in defense against pathogens and enhance plant growth and development. They can also bolster plant immunity and stress resistance, improve soil quality, increase soil fertility and contribute to vegetation recovery and ecosystem stability [95].

In this review, we comprehensively summarized the chemical structure types, biosyntheses, bioactivities, sources, and distributions of secondary metabolites isolated from *Cladosporium* in the period from January 2000 to December 2022. The literature survey indicates that the genus *Cladosporium*, especially marine-derived *Cladosporium*, has great potential as a producer to generate abundant and diverse new bioactive natural products. Some potent antibacterial and cytotoxic compounds isolated from *Cladosporium* have the potential to be developed into new drugs. Additionally, all the natural products isolated from *Cladosporium* provide a structural foundation for drug design.

## Figures and Tables

**Figure 1 ijms-25-01652-f001:**
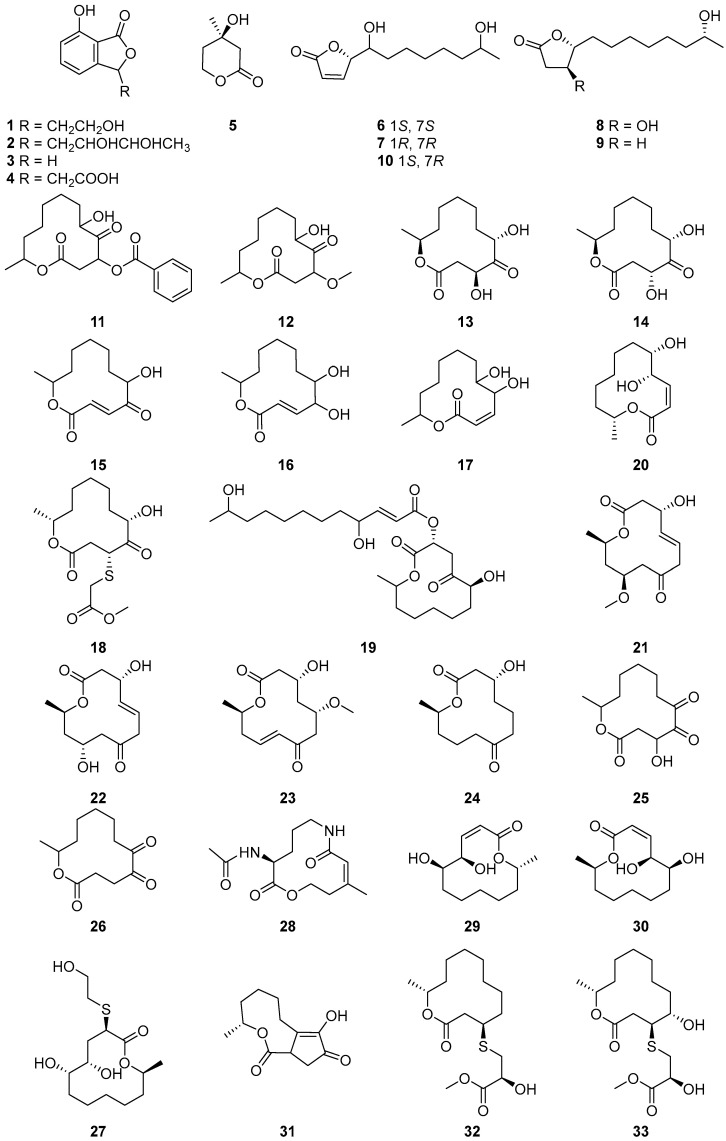
Structures of compounds **1**–**33** [22,32,33,34,35,36,37,38,39,40,41,42].

**Figure 2 ijms-25-01652-f002:**
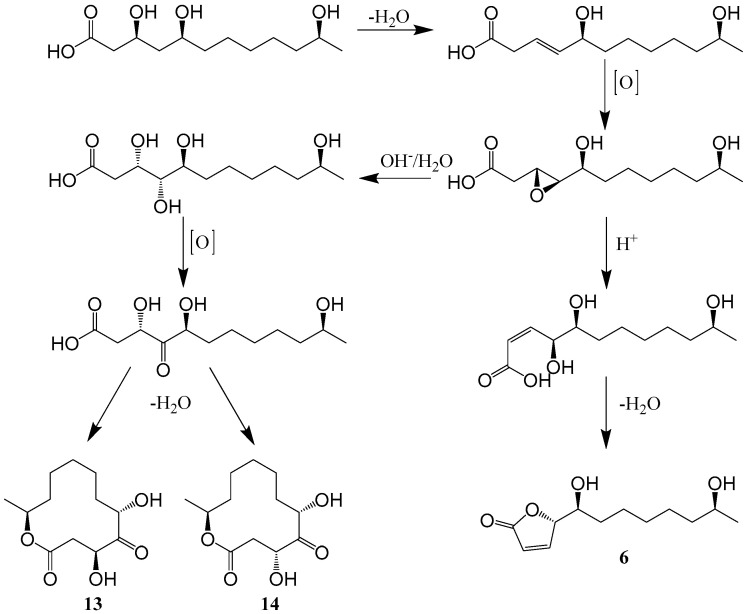
Proposed biosynthesis pathway for compounds **6**, **13** and **14** [34].

**Figure 3 ijms-25-01652-f003:**
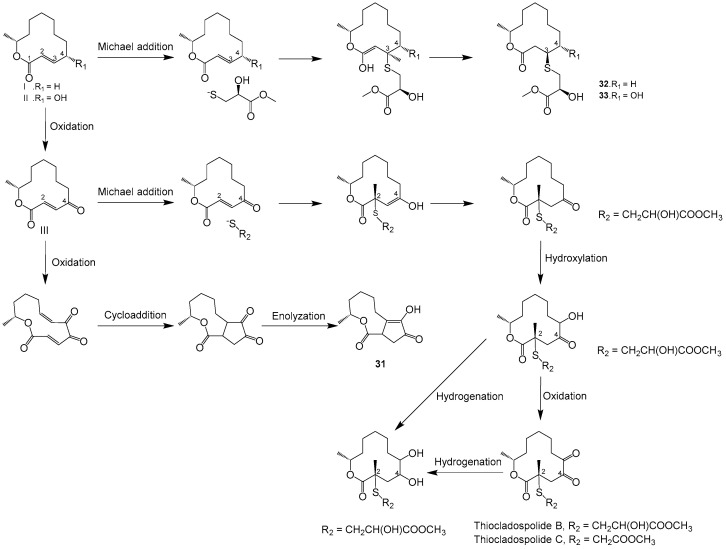
Proposed biosynthesis pathway for compounds **31**–**33** [32].

**Figure 4 ijms-25-01652-f004:**
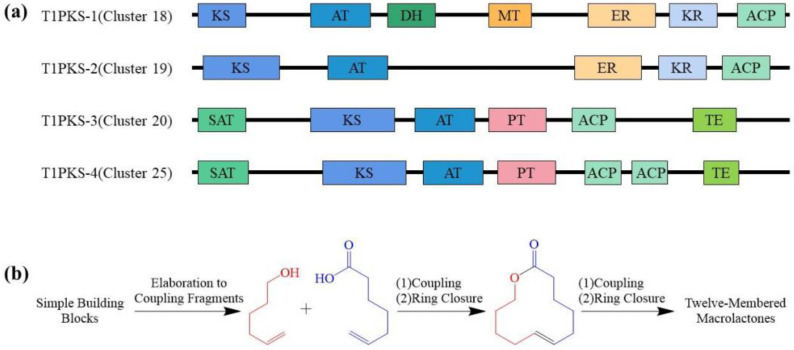
(**a**) Gene clusters of secondary metabolic synthesis genes of *Cladosporium*. KS: Ketoacyl synthase; AT: Acyl transferase; DH: Dehydratase; MT: Methyltransferase; ER: Enoyl reductase; KR: Ketolreductase; ACP: Acyl carrier protein; SAT: Starting unit acyltransferase; PT: Product template; TE: Thioesterase [46,47,48,49]. (**b**) Proposed biosynthesis of the 12-membered macrolactones [44,45].

**Figure 5 ijms-25-01652-f005:**
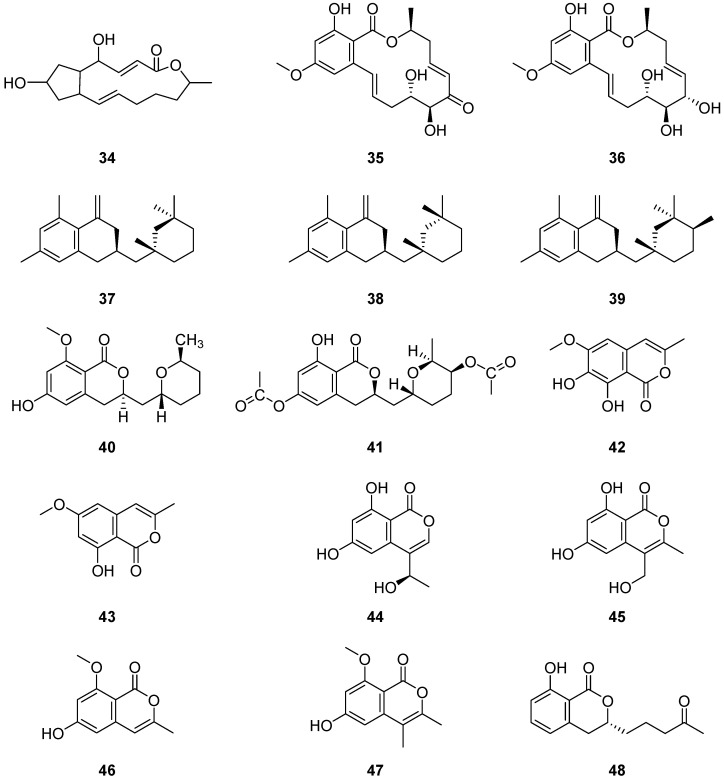
Structures of compounds **34**–**48** [50,53,54,55,56].

**Figure 6 ijms-25-01652-f006:**
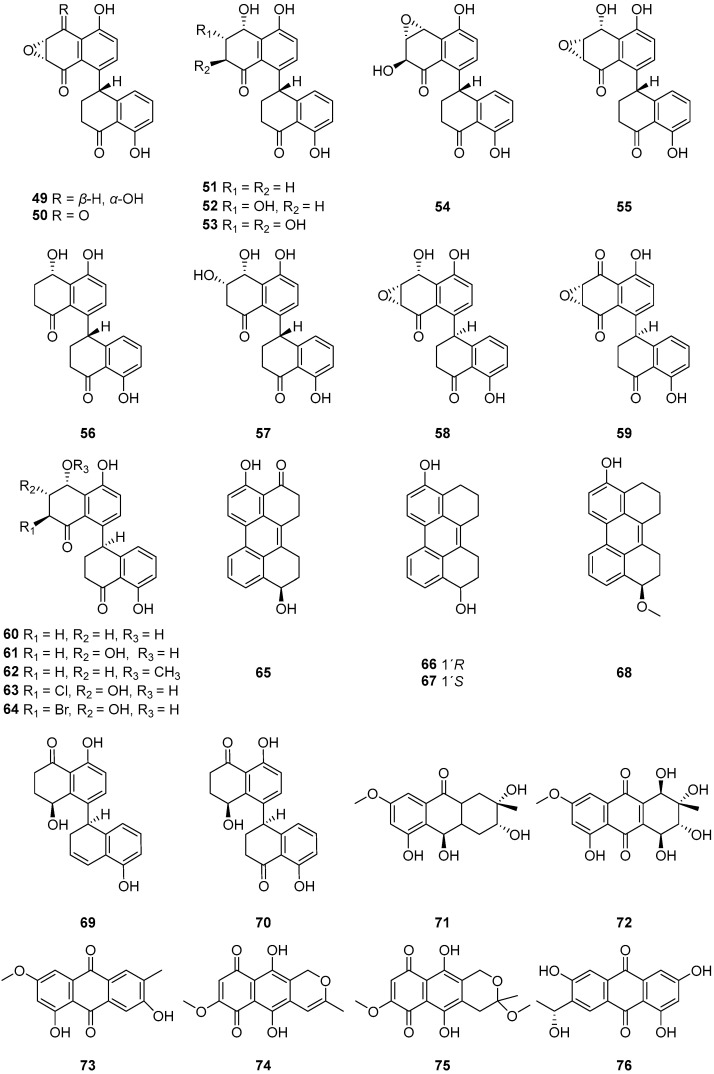
Structures of compounds **49**–**76** [19,32,57,58,59,60,61].

**Figure 7 ijms-25-01652-f007:**
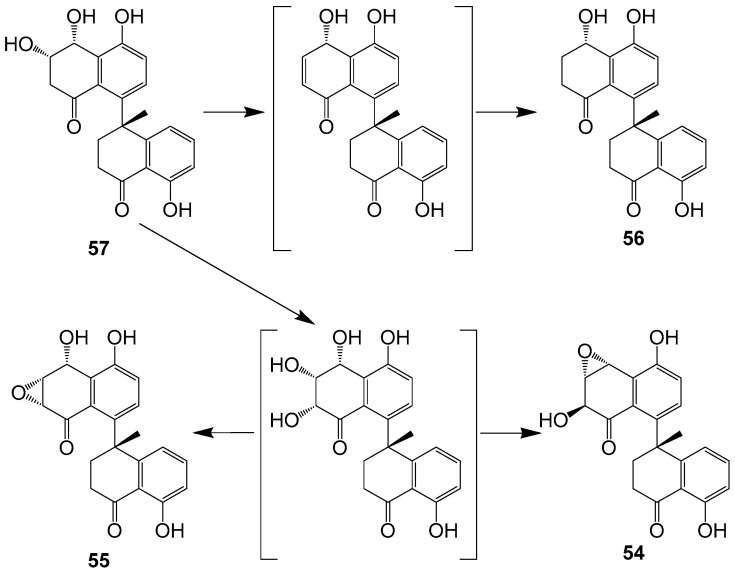
Proposed biosynthesis mechanism for compounds **54**–**57** [19].

**Figure 8 ijms-25-01652-f008:**
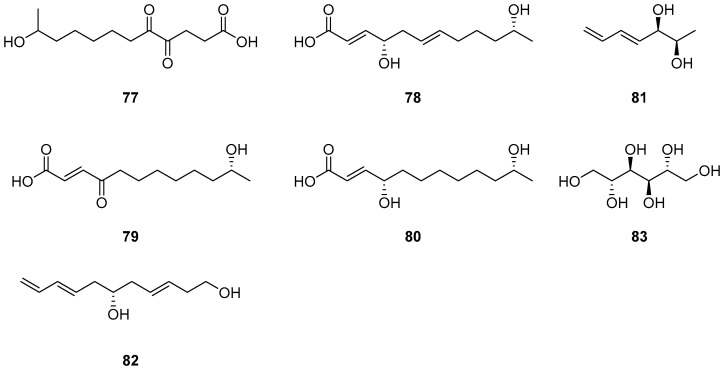
Structures of compounds **77**–**83** [20,25,38,55].

**Figure 9 ijms-25-01652-f009:**
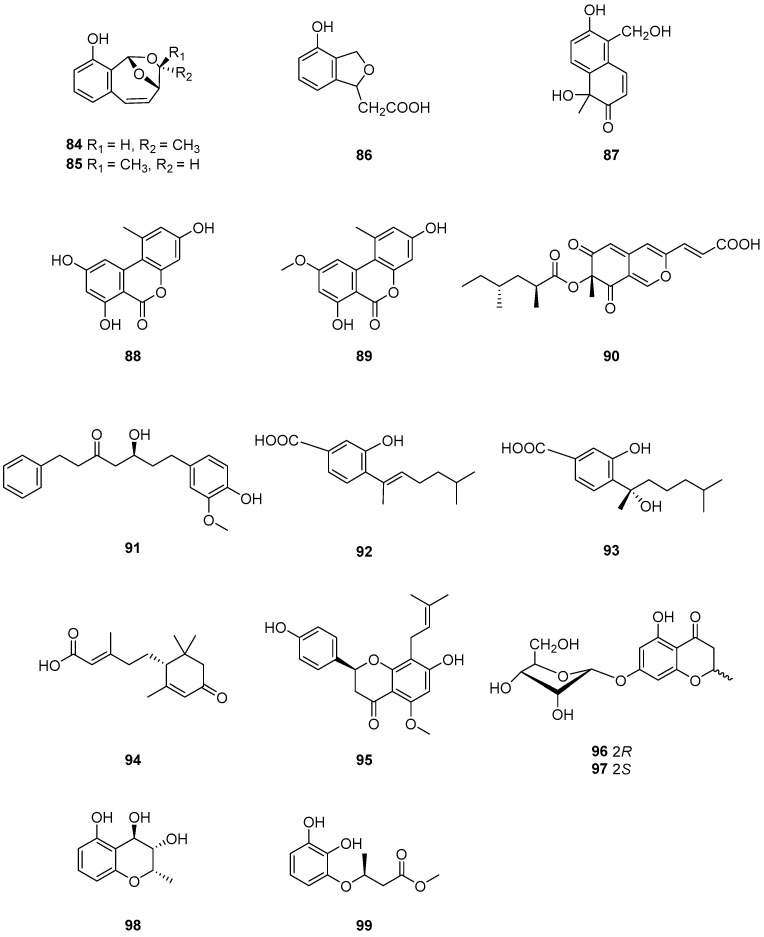
Structures of compounds **84**–**99** [20,32,53,64,65,66,67].

**Figure 10 ijms-25-01652-f010:**
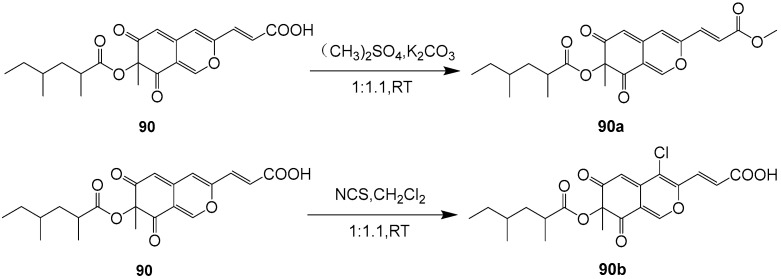
Chemical synthetic pathways of compound **90** derivatives [53].

**Figure 11 ijms-25-01652-f011:**
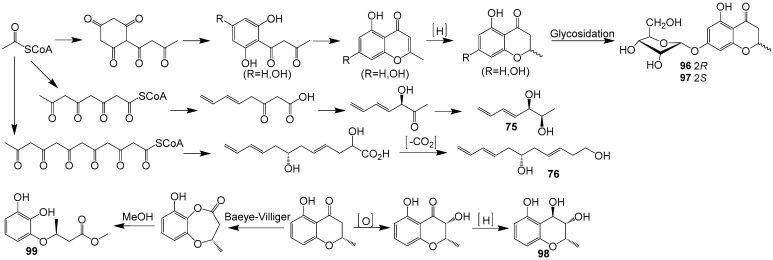
Proposed biosynthesis pathway for compounds **75**–**76** and **96**–**99** [20].

**Figure 12 ijms-25-01652-f012:**
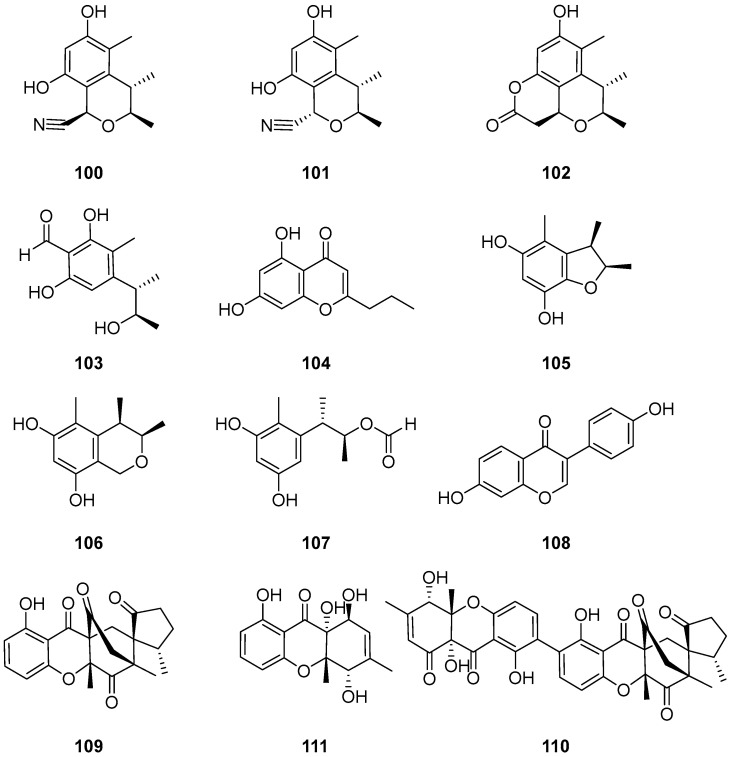
Structures of compounds **100**–**111** [56,66,70,71].

**Figure 13 ijms-25-01652-f013:**
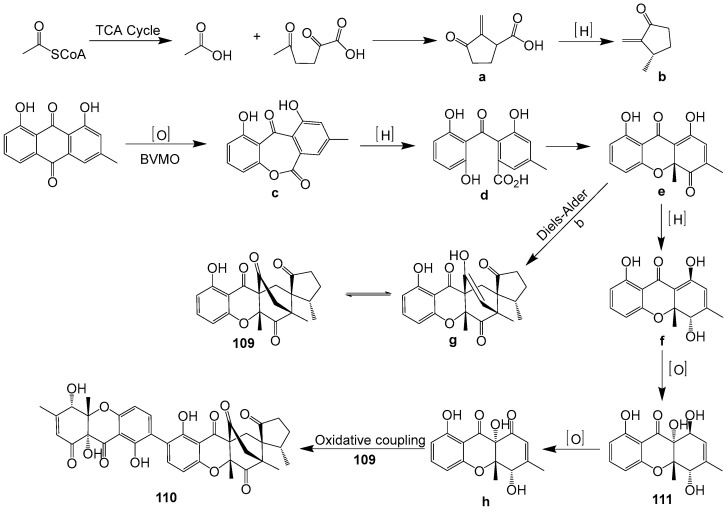
Proposed biosynthesis pathway for compounds **109**–**111** [71].

**Figure 15 ijms-25-01652-f015:**
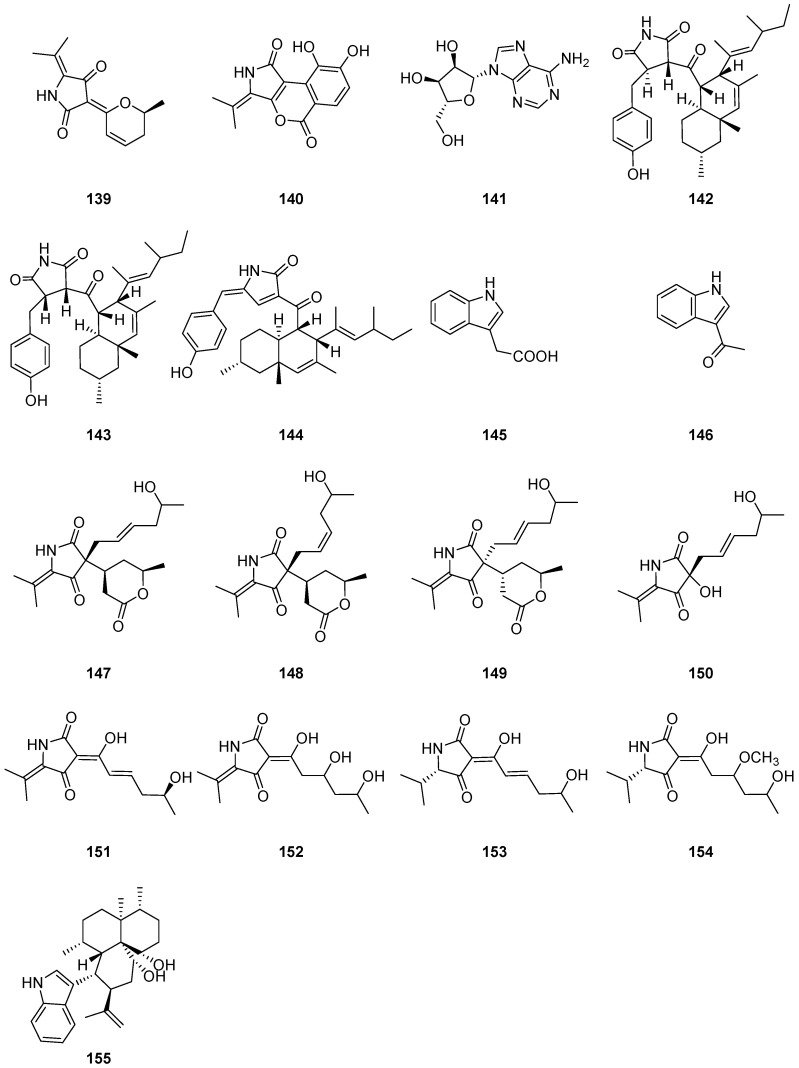
Structures of compounds **139**–**155** [55,61,66,68,69,77,78,79].

**Figure 16 ijms-25-01652-f016:**
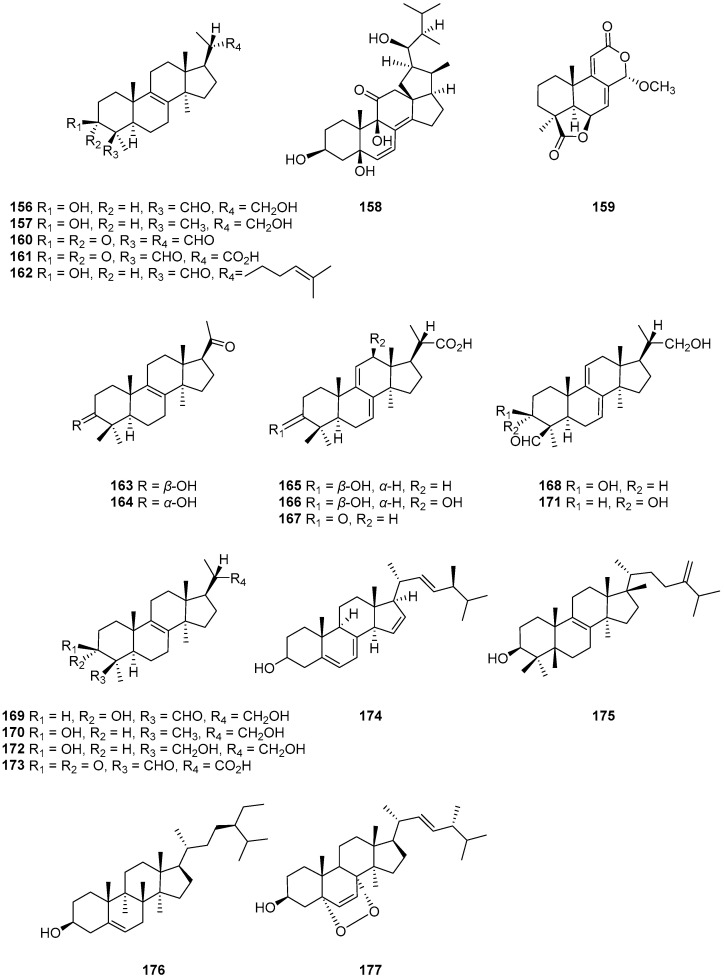
Structures of compounds **156**–**177** [13,65,80,81].

**Figure 17 ijms-25-01652-f017:**
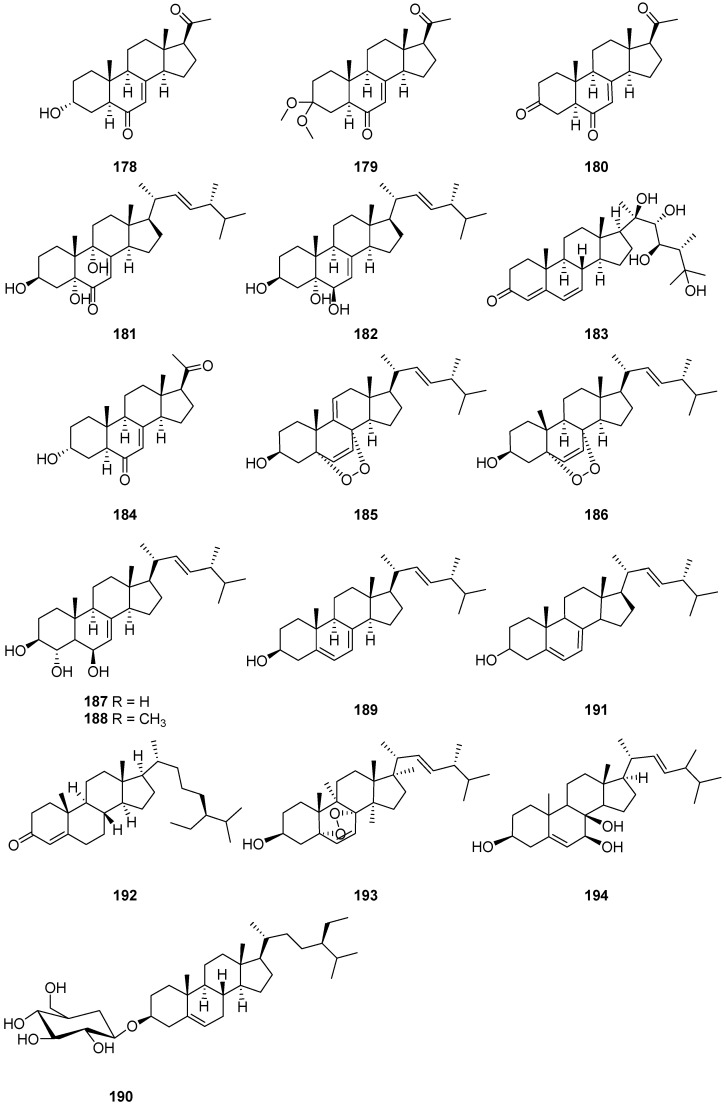
Structures of compounds **178**–**194** [21,28,41,55,66].

**Figure 18 ijms-25-01652-f018:**
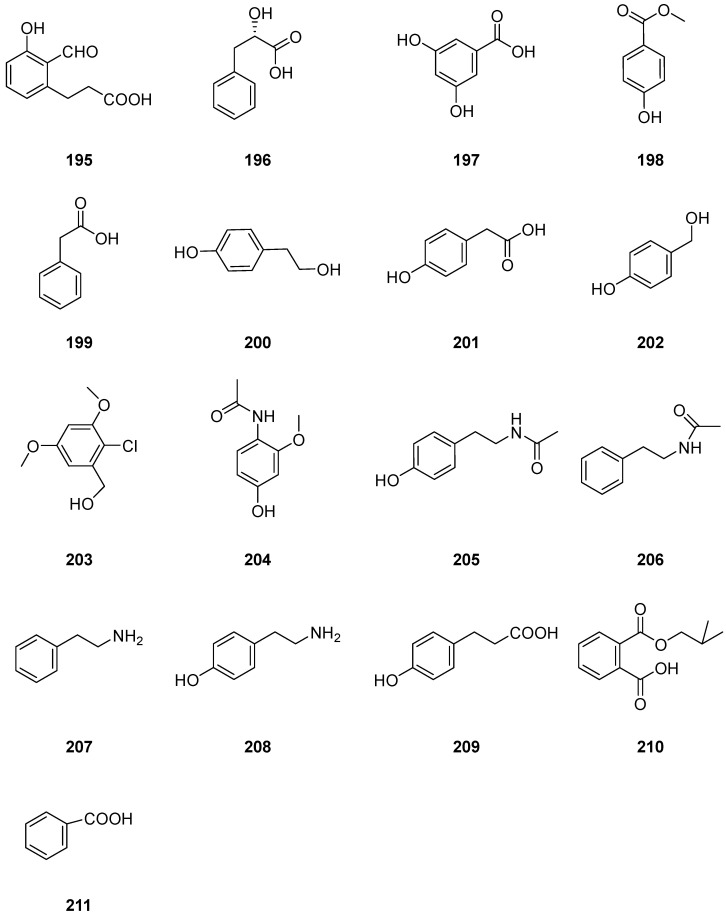
Structures of compounds **195**–**211** [32,33,56,65,66].

**Figure 19 ijms-25-01652-f019:**
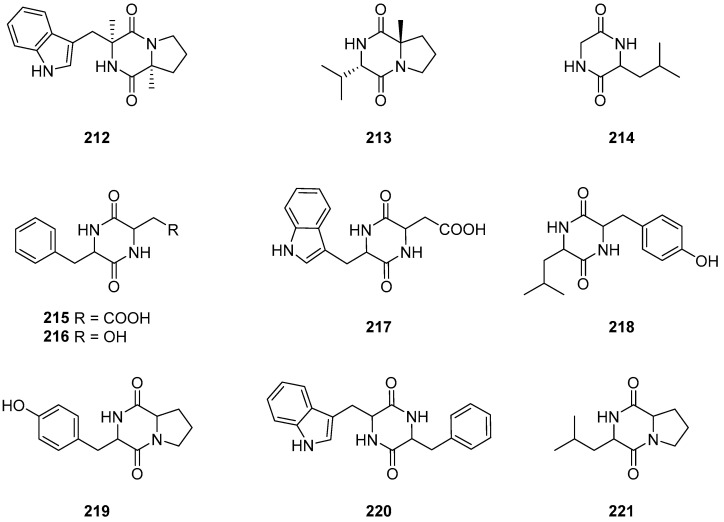
Structures of compounds **212**–**221** [28,33,66].

**Figure 20 ijms-25-01652-f020:**
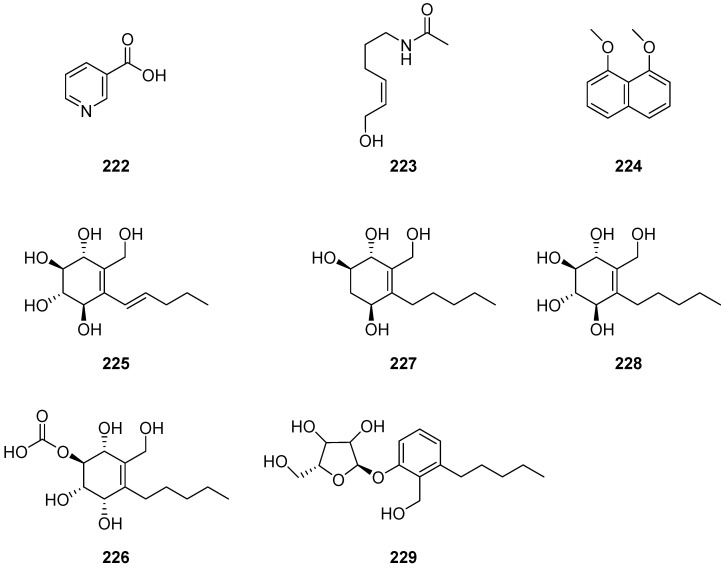
Structures of compounds **222**–**229** [23,33,56].

**Figure 21 ijms-25-01652-f021:**
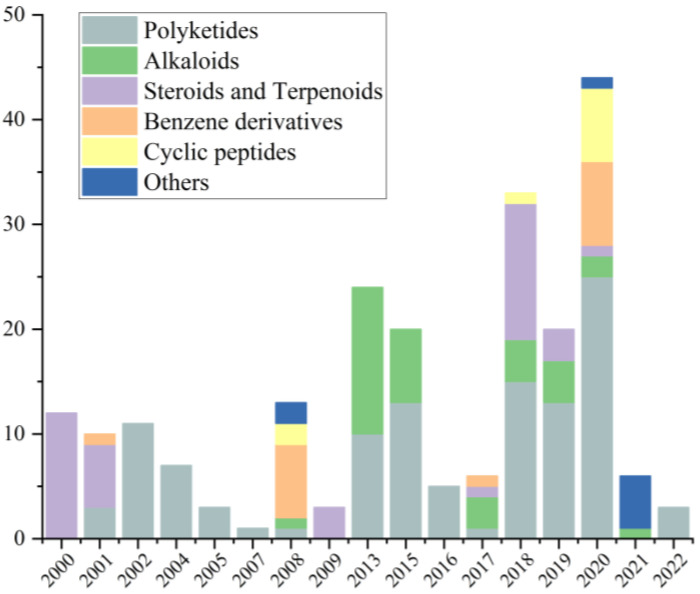
Structural types of compounds isolated from *Cladosporium* over different years.

**Figure 22 ijms-25-01652-f022:**
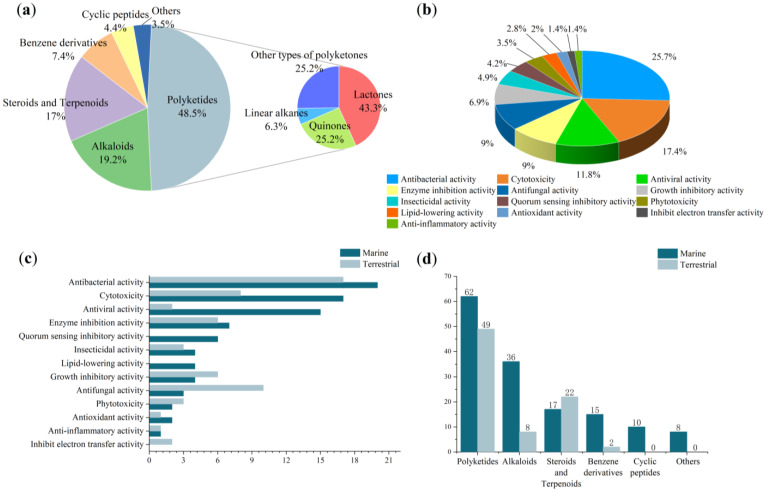
(**a**) Structural types of compounds isolated from *Cladosporium* from January 2000 to December 2022; (**b**) Bioactivities of natural products isolated from *Cladosporium* discovered from January 2000 to December 2022; (**c**) Activities of compounds isolated from marine and terrestrial *Cladosporium* from January 2000 to December 2022; (**d**) Structural types of compounds of marine and terrestrial isolated from *Cladosporium* from January 2000 to December 2022.

**Figure 23 ijms-25-01652-f023:**
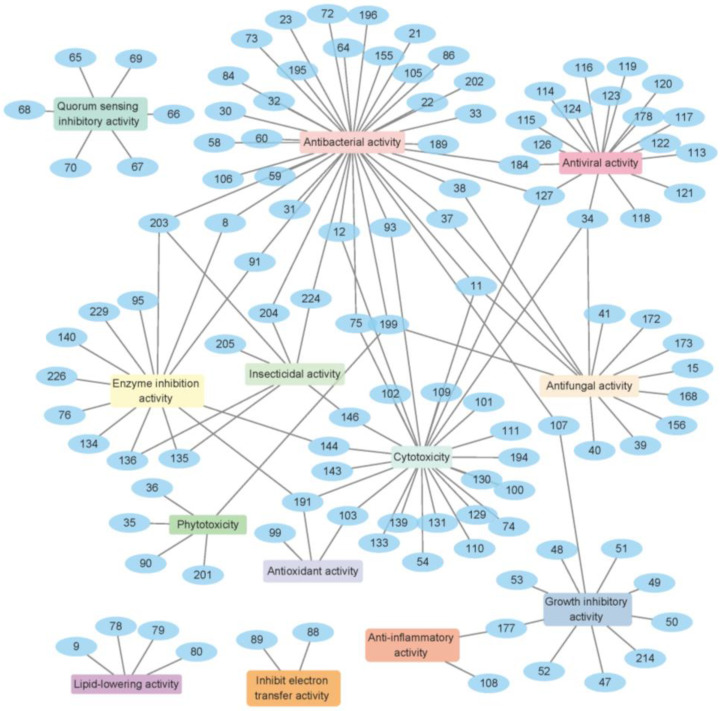
Bioactive molecular network of natural products isolated from *Cladosporium* from January 2000 to December 2022.

**Table 1 ijms-25-01652-t001:** Compounds isolated from *Cladosporium* from January 2000 to December 2022.

Types	Comps.	Sources	Distribution	Years	Refs.
Polyketides	**1**–**4**	*Cladosporium* sp. NRRL 29097	Malette	2002	[32]
	**5**	*Cladosporium* sp. N5	Jiangsu China	2008	[33]
	**6**	Sponge *Niphatesrowi* sp. derived *Cladosporium* sp. DQ100370	the Red Sea	2005	[34]
	**7**	Gorgonian *Anthogorgia ochracea* (GXWZ-07)-derived *Cladosporium* sp. RA07-1 (GenBank No. KP720581)	the South China Sea	2015	[35]
	**8**–**10**	Soft coral-derived fungus *Cladosporium* sp. TZP-29 (GenBank No. KR817674)		2019	[36]
	**11**–**12**	Marine-derived fungus *Cladosporium* sp. L037	Okinawa Island	2004	[37]
	**13**–**14**	Sponge *Niphatesrowi* sp. derived *Cladosporium* sp. DQ100370	the Red Sea	2005	[34]
	**15**–**17**	Sponge-derived *Cladosporium* sp. FT-0012	Pohnpei island	2001	[22]
	**18**–**19**	Plant *Rhizophora stylosa*-derived endophytic fungus *Cladosporium* sp. IFB3lp-2	Hainan China	2013	[38]
	**20**–**24**	Gorgonian *Anthogorgia ochracea* (GXWZ-07)-derived *Cladosporium* sp. RA07-1 (GenBank No. KP720581)	the South China Sea	2015	[35]
	**25**–**26**	plant *Callistemon viminalis*-derived fungus *Cladosporium* sp. A801 (GenBank No.MF138133)	Guangzhou China	2019	[39]
	**27**–**29**	Mangrove endophytic fungus *Cladosporium* sp. SCNU-F0001 (GenBank NO. NG062723.1)	Guangzhou China	2019	[40]
	**30**	Endophytic fungus *Cladosporium* sp. IS384 (GenBank NO. KU158172)	Sichuan China	2019	[41]
	**31**–**33**	Marine mangrove-derived endophytic fungus, *Cladosporium cladosporioides* MA-299 (GenBank No. MH822624)	Hainan China	2020	[42]
	**34**	*Cladosporium* sp. I(R)9-2	Nanjing China	2007	[50]
	**35**–**36**	*Cladosporium oxysporum* DH14 (GenBank No. JN887339.1)	Jiangsu China	2016	[53]
	**37**–**41**	Endophytic fungus *Cladosporium cladosporioides*	Tifton	2013	[54]
	**42**–**43**	Plant *Ammopiptanthus mongolicus*-derived fungus *Cladosporium cladosporioides*		2019	[55]
	**44**–**48**	Plant *Ceriops tagal*-derived fungus *Cladosporium* sp. JS1-2	Hainan China	2020	[56]
	**49**–**53**	*Cladosporium tenuissimum* ITT21	Tuscany Italy	2004	[57]
	**54**–**57**	Mangrove plant *Kandelia candel*-derived *Cladosporium* sp. KcFL6′	Guangzhou China	2015	[19]
	**58**–**64**	*Cladosporium* sp. TMPU1621	Okinawa Japan	2018	[58]
	**65**–**70**	*Cladosporium* sp. KFD33 (GenBank No.MN737504)	Hainan China	2020	[59]
	**71**–**73**	*Cladosporium* sp. NRRL 29097	Red River	2002	[32]
	**74**–**75**	*Cladosporium* sp. RSBE-3		2016	[60]
	**76**	Mangrove-derived fungus *Cladosporium* sp. HNWSW-1 (GenBank No. MH 535968)	Hainan China	2019	[61]
	**77**	Plant *Rhizophora stylosa*-derived endophytic fungus *Cladosporium* sp. IFB3lp-2	Hainan China	2013	[38]
	**78**–**80**	Soft coral-derived fungus *Cladosporium* sp. TZP-29 (GenBank NO.KR817674)		2015	[25]
	**81**–**82**	*Cladosporium* sp. OUCMDZ-302	Hainan China	2018	[20]
	**83**	Plant *Ammopiptanthus mongolicus*-derived fungus *Cladosporium cladosporioides*		2019	[55]
	**84**–**87**	*Cladosporium* sp. NRRL 29097	Red River	2002	[31]
	**88**–**89**	Plant endangered *Chrysosplenium carnosum*-derived fungus *Cladosporium* sp. J6 (GenBank NO.KR492688)	Tibet China	2013	[64]
	**90**	*Cladosporium oxysporum* DH14 (GenBank NO. JN887339.1)		2016	[53]
	**91**	*Cladosporium cladosporioides* JG-12	Hainan China	2017	[65]
	**92**–**93**	Marine sediment-derived *Cladosporium* sp. NJF6	Antarctica	2020	[66]
	**94**	Marine-derived *Cladosporium* sp. OUCMDZ-1635 (GenBank No. KT336457)	Xisha Islands of China	2018	[68]
	**95**	Marine sponge-derived *Cladosporium* sp. TPU1507	Manado, Indonesia	2018	[69]
	**96**–**99**	*Cladosporium* sp. OUCMDZ-302	Hainan, China	2018	[20]
	**100**–**103**	Deep sea-derived fungus *Cladosporium* sp. SCSIO z015 (GenBank No. KX258800)	Okinawa Trough	2020	[70]
	**104**–**107**	*Cladosporium* sp. JS1-2	Hainan China	2020	[56]
	**108**	Marine sediment-derived *Cladosporium* sp. NJF6	Antarctica	2020	[66]
	**109**–**111**	*Cladosporium* sp. (GenBank No. QH071013)	Qinghai China	2022	[71]
Alkaloids	**112**	*Cladosporium* sp. N5(GenBank No. EF424419)	Lianyungang China	2008	[33]
	**113**–**126**	Mangrove-derived fungus *Cladosporium* sp. PJX-41 (GenBank No. KC589122)	Guangzhou China	2013	[26]
	**127**–**128**	Plant endangered *Chrysosplenium carnosum*-derived fungus *Cladosporium* sp. J6 (GenBank NO. KR492688)	Tibet China	2015	[63]
	**129**–**131**	Marine sediment-derived *Cladosporium* sp.	Zhejiang China	2015	[76]
	**132**	Plant endangered *Chrysosplenium carnosum*-derived fungus *Cladosporium* sp. J6 (GenBank NO. KR492688)	Tibet China	2015	[63]
	**133**	Gorgonian *Anthogorgia ochracea* (GXWZ-07)-derived *Cladosporium* sp.RA07-1 (GenBank NO. KP720581)	the South China Sea	2015	[35]
	**134**–**136**	*Cladosporium cladosporioides* JG-12	Hainan China	2017	[65]
	**137**–**138**	Marine sponge *Callyspongia* sp. derived fungus *Cladosporium* sp. scsio41007 (GenBank NO.MF188197)	Guangzhou China	2018	[28]
	**139**	Marine-derived fungus *Cladosporium* sp. OUCMDZ-1635 (GenBank No. KT336457)	Xisha Islands of China	2018	[68]
	**140**	Marine sponge-derived *Cladosporium* sp. TPU1507	Indonesian	2018	[69]
	**141**	Plant *Ammopiptanthus mongolicus*-derived fungus *Cladosporium cladosporioides*		2019	[55]
	**142**–**144**	Mangrove-derived fungus *Cladosporium* sp. HNWSW-1 (GenBank access No. MH 535968)	Hainan China	2019	[61]
	**145**–**146**	Marine sediment-derived *Cladosporium* sp. NJF6	Antarctica	2020	[66]
	**147**–**154**	Deep sea sediment-derived *Cladosporium* sp. SCSIO z0025	Okinawa	2018	[78]
	**155**	*Cladosporium* sp. JNU17DTH12-9-01 (GenBank no. MK994007)		2021	[79]
Steroids and Terpenoids	**156**–**167**	*Cladosporium* sp. IFM49189		2000	[13]
	**168**–**173**	*Cladosporium* sp. IFM49189		2001	[80]
	**174**–**176**	Marine mangrove-derived fungus *Cladosporium cladosporioides*	Guangzhou China	2009	[81]
	**177**	*Cladosporium cladosporioides* JG-12	Hainan China	2017	[65]
	**178**–**183**	Marine sponge-derived fungus *Cladosporium* sp. SCSIO41007	Guangzhou China	2018	[28]
	**184**–**190**	Gorgonian-derived fungus *Cladosporium* sp. WZ-2008-0042	the South China Sea	2018	[21]
	**191**	Endophytic fungus *Cladosporium* sp. IS384	Sichuan China	2019	[41]
	**192**–**193**	Plant *Ammopiptanthus mongolicus*-derived fungus *Cladosporium cladosporioides*		2019	[55]
	**194**	Marine sediment-derived *Cladosporium* sp. NJF4		2020	[66]
Benzene derivatives	**195**	*Cladosporium* sp. NRRL 29097	Red River	2001	[32]
	**196**–**202**	*Cladosporium* sp. N5 (GenBank No. EF424419)	Lianyungang China	2008	[33]
	**203**	*Cladosporium cladosporioides* JG-12	Hainan China	2017	[65]
	**204**–**205**	*Cladosporium* sp. JS1-2	Hainan China	2020	[56]
	**206**–**211**	Marine sediment-derived *Cladosporium* sp. NJF6	Antarctica	2020	[66]
Cyclic peptides	**212**–**213**	*Cladosporium* sp. N5 (GenBank No. EF424419)	Lianyungang China	2008	[33]
	**214**	Marine sponge *Callyspongia* sp. derived fungus *Cladosporium* sp. scsio41007 (GenBank NO.MF188197)	Guangzhou China	2018	[28]
	**215**–**221**	Marine sediment-derived *Cladosporium* sp. NJF4	Antarctica	2020	[66]
Others	**222**–**223**	*Cladosporium* sp. N5	Lianyungang China	2008	[33]
	**224**	*Cladosporium* sp. JS1-2	Hainan China	2020	[56]
	**225**–**229**	Mangrove-derived fungus *Cladosporium* sp. JJM22 (GenBank No.MF593626)	Hainan China	2021	[23]

**Table 2 ijms-25-01652-t002:** Antibacterial activities of compounds isolated from *Cladosporium* during 2000–2022.

Strains	Comps.	Values (MIC)	Values of Positive Controls (MIC)	Pros	Cons
*Staphylococcus aureus* (MRSA)	**8** (μg/mL) [36]	8.0	1.0		Moderate inhibitory activity
*Micrococcus luteus*	**11**–**12** (μg/mL) [37]	16.7			Moderate activities
*Bacillus cereus*	**21**–**23** (μM) [35]	12.5/25.0/6.25	1.56	Broad-spectrum antibacterial activity; compounds **21** and **22** exhibited the strongest activities against *T. halophilus*	Moderate activity
*Tetrag enococcus halophilus*		3.13/3.13/25.0	1.56		
*S. epidermidis*		6.25/25.0/25.0	0.78		
MRSA		6.25/25.0/12.5	0.39		
*Escherichia coli*		12.5/12.5/25.0	1.56		
*Pseudomonas putida*		12.5/25.0/6.25	0.39		
*Nocardia brasiliensis*		6.25/12.5/25.0	0.78		
*Vibrio parahaemolyticus*		12.5/25.0/25.0	1.56		
*Enterococcus faecalis* ATCC 29212	**30** (μg/mL) [41]	0.31	20.83	Strong activity	
*Edwardsiella tarda*	**31**–**33** (μg/mL) [42]	1.0/2.0/2.0	0.5	Strong activities	
*V. anguillarum*		2.0/2.0/4.0	1.0		
MRSA ATCC43300	**58**/**59**/**60**/**64** (μg/mL) [58]	25/3.13/25/25	0.78	Compound **59** displayed strong activity of MRSA	Compounds **58** and **60** showed weak activities
MRSA ATCC700698		25/12.5/50/25	1.56		
*B. subtilis*	**72**–**73** (inhibition zone) (mm) [32]	33/23			Moderate activities
MRSA		31/20			
MRSA	**75** (inhibition zone) (mm) [60]	27	32	Strong and broad-spectrum activity	
*E. coli*		25	30		
*Pseudomonas aeruginosa*		24	30		
*B. megaterium*		22	32		
MRSA	**84** (inhibition zone) (mm) [32]	13			Moderate activity
*B. subtilis*	**86** (inhibition zone) (mm) [32]	8			Weak activity
*Ralstonia solanacearum*	**91** (inhibition zone) (mm) [65]	6.29 ± 0.10	25.16 ± 0.06		Weak activity
MRSA		6.45 ± 0.11	17.62 ± 0.08		
*B. subtilis*	**93** (μM) [66]	2.50	1.25	Broad-spectrum activity	
*Sarcina lutea*		2.50	2.50		
*E. coli*		5.00	0.625		
*M. tetragenus*		20.0	0.312		
*V. parahaemolyticus*		10.0	0.160		
*V. anguillarum*		5.0	0.160		
MRSA	**105**–**107** (μg/mL) [56]	12.5	3.12		Moderate activities
MRSA 209P	**155** (μg/mL) [79]	4	0.13		Weak activity
*Candida albicans* FIM709		16	0.06		
*Shigella dysenteriae*	**189** (μM) [21]	3.13			Moderate activity
*B. subtilis*	**195** (inhibition zone) (mm) [32]	22			Weak activity
*C. albicans*	**203** (inhibition zone) (mm) [65]	7.43 ± 0.12	12.00 ± 0.09	Strong activity	
MRSA	**204**/**224** (μg/mL) [56]	12.5/12.5	3.12		Moderate activities

**Table 3 ijms-25-01652-t003:** Cytotoxicity of compounds isolated from *Cladosporium* during 2000–2022.

Cells	Comps.	Values (IC_50_)	Values of Positive Controls (IC_50_)	Pros	Cons
L1210	**11**–**12** (μg/mL) [37]	0.13/0.81		Strong activities	
MCF–7	**34** (μM) [50]	0.2		Strong and broad-spectrum cytotoxicity	
A549		0.3			
HCT116		0.5			
786-O		0.7			
PC3		1.5			
K562	**54** (μM) [19]	14.3	0.24		Weak activity
A549		15.7	0.05		
Huh-7		29.9	0.08		
H1975		40.6	0.09		
MCF-7		21.3	0.78		
U937		10.5	0.06		
BGC823		17.0	0.09		
HL60		10.1	0.09		
Hela		53.7	0.11		
MOLT-4		14.6	0.03		
K-562	**74**–**75** (μg/mL) [60]	3.97/3.58	12.0	Potential cytotoxicities against human leukemia cells (K-562)	
HL-60	**93** (μM) [66]	>50			Weak activity
A-549		>50			
Brine shrimp naupalii	**100**–**103** (μM) [70]	72.0/81.7/49.9/81.4	21.2		Moderate activities
MB49	**109**–**110** (μM) [71]	13.9 ± 2.5/41.7 ± 7.5	2.7 ± 0.6	Broad-spectrum cytotoxicity	Weak activities
J82		25.0 ± 6.1/24.7 ± 4.4	0.6 ± 0.1		
4T1		38.7 ± 4.2/27.5 ± 2.8	4.5 ± 1.6		
Huh7		24.3 ± 3.5/46.4 ± 9.3	5.1 ± 1.2		
MCF-7	**127** (μM) [63]	20		Broad-spectrum cytotoxicity	Moderate activity
A549		15			
HT-29		10			
HepG2		10			
HepG2	**129**–**131** (μg/mL) [76]	21/42/48			Moderate activities
HeLa	**133** (μM) [35]	0.76		Potent cytotoxicity against a series of cancer cell lines and broad-spectrum cytotoxicity	
P388		1.35			
HT-29		2.48			
A549		3.11			
MCF-7	**139** (μM) [68]	18.7	0.67	Broad-spectrum cytotoxicity	Moderate activity
HeLa		19.1	0.32		
HCT-116		17.9	0.21		
HL-60		9.1	0.02		
BEL-7042	**143**–**144** (μM) [60]	29.4 ± 0.35/26.7 ± 1.1	11.9 ± 0.37	Broad-spectrum cytotoxicities	Moderate activities
K562		25.6 ± 0.47/-	14.2 ± 0.66		
SGC-7901		41.7 ± 0.71/-	6.66 ± 0.2		
Hela		-/14.9 ± 0.21	11.5 ± 0.18		
Vero	**146** (μM) [66]	87			Weak activity
HeLa	**191** (μM) [41]	22			Weak activity
HeLa	**194** (μM) [66]	14.1			Moderate activity

**Table 4 ijms-25-01652-t004:** Other activities of compounds isolated from *Cladosporium* during 2000–2022.

Bioactivities	Cells/Stains/Enzyme	Comps.	Values	Values of Positive Controls	Pros	Cons
Antiviral activity	HBV	**34** (μM) [50]	0.5		Strong and broad-spectrum antiviral activity	
	HIV-1		1.0			
	HCMV		1.5			
	IAV		0.5			
	H1N1	**113**–**126** (μM) [26]	80–150	87	Compounds **115**, **118**, **120**–**122** and **124** showed noteworthy antiviral activities	Compounds **113**, **114**, **116**, **117**, **119**, **123** and **126** showed weak antiviral activities
	HBV	**127** (μM) [63]	5		Broad-spectrum antiviral activity	
	HIV-1		5			
	HCMV		5			
	IAV		5			
	H3N2	**178** (μM) [28]	16.2	34.0		Weak activity
	RSV	**184** (mM) [21]	0.12	0.08	Potent antiviral activity	
Antifungal activity	*Candida albicans*	**11** (μg/mL) [37]	16.7		Strong and broad-spectrum antifungal activity	
	*Cryptococcus neoformans*		8.4			
	*Aspergillus niger*		16.7			
	*Neurospora crassa*		8.4			
	*Pyricularia oryzae*	**15** (μg/mL) [22]	0.15		Strong antifungal activity	
	*Muco rracemosus*		29			
	*A. niger*	**34** (μg/mL) [50]	0.97		Strong activity	
	*C. albicans*		1.9			
	*Trichophyton rubrum*		1.9			
	*Colletotrichum acutatum*	**37**–**41** (tested at 30 µM) (%) [54]	92.7/38.3/-/-/-	99.7	Compound **37** showed broad-spectrum antifungal activity, and the activity is significant	
	*Co. fragariae*		90.1/50.4/-/-/-	99.6		
	*Co. gloeosporioides*		95.4/60.2/-/-/-	96.1		
	*Phomopsis viticola*		79.9/83.0/53.9/35.1/79.4	94.2		
	*P. obscurans*		22.1/22.5/25.6/-/10.3	96.2		
	*A. Fumigatus* IFM 4942/40849/41375/41382/46075/47064/47078/47031/47032	**156** (μg/disc) [13]	0.5–4		Strong and exhibited specific antifungal activity toward *A. fumigatus*	
	*A. fumigatus* IFM 4942/40819/41375/46075/47064/47078	**168**, **172**–**173** (μg/disc) [80]	9–17			Compounds **172** and **173** retained weak antifungal activities against *A. fumigatus*
	*C. albicans*	**199** (mM) [33]	1.5		Broad-spectrum antifungal activity	Weak activity
	*Pseudomonas aeruginosa*		3.1			
	*Staphylococcus aureus*		0.78			
Enzyme inhibition activity	Acetylcholinesterase	**8** (μM) [36]	40.26		Potent inhibitory activity against acetylcholinesterase	
	α-glycosidase	**76** (μM) [61]	49.3 ± 10.6	275.7 ± 2.7	Strong activity	
	Acetylcholinesterase	**91** (tested at 1 g/L) (%) [65]	23.54 ± 0.70	77.43 ± 1.47		Weak activity
	PTP1B	**95** (μM) [69]	11	0.9		Moderate inhibitory activity
	Acetylcholinesterase	**134**–**136** (tested at 1 g/L) (%) [65]	37.20 ± 1.31/26.94 ± 5.64/26.35 ± 1.55	77.43 ± 1.47		Weak activity
	PTP1B	**140** (μM) [69]	48	0.9	Strong activity	
	TCPTP		54	0.8		
	α-glycosidase	**144** (μM) [61]	78.2 ± 2.1	275.7 ± 2.7	Strong activity	
	Acetylcholinesterase	**203** (tested at 1 g/L) (%) [65]	25.43 ± 1.08	77.43 ± 1.47		Weak activity
	*α*-glucosidase	**226**, **229** (μM) [23]	2.95/2.05	2.35	Potent inhibitory activities against *α*-glucosidase	
Growth inhibitory activity	Growth inhibition activity against newly hatched larvae of *Helicoverpa armigera* Hubner	**47**–**48**, **107**, **204**–**205**, **224** (μg/mL) [56]	100/100/100/100/100/100	50		Moderate inhibitory activities
	Inhibition of *Uromyces appendiculatus* urediniospore germination	**49**–**53** (tested at 100 ppm) (%) [57]	84.2/100/77.6/69.4/74.8	20.9	Strong inhibitory activities	
	Anti-inflammatory activity	**177** (μM) [65]	27.9	61.7	Potent anti-inflammatory activity	
Insecticidal activity	*Panagrellus redivivus*	**135**–**136**, **203** (tested at 2.5 g/L) (%) [65]	78.2 ± 0.7/80.7 ± 0.4/89.6 ± 0.9	37.2 ± 0.3	Significant and comparable to that of the positive control	
	*Trypanosoma cruzi*	**146** (μM) [66]	26.9			Moderate activity
Quorum sensing inhibitory activity	*Chromobacterium violaceum* CV026	**65**–**70** (μg/well) [59]	30/30/20/30/20/30		Strong activities	
Phytotoxicity	*Amaranthus* *retroflexus* L.	**35**–**36**, **90** (μg/mL) [53]	4.80/8.16/4.51	1.95	Potent phytotoxic activities against the radicle growth of *Amaranthus retroflexus* L.	
	*Porphyra tenera*	**199**, **201** (mm) [33]	1.57 ± 0.06/0.95 ± 0.04	0.98 ± 0.05	Strong activities	
Lipid-lowering activity	HepG2 hepatocytes	**9**, **78**–**80** (μM) [36]	8.3/12.1/8.4/13.1	8.3	Potent lipid-lowering activities in HepG2 hepatocytes	
Antioxidant activity	DPPH free radical scavenging	**99** (μM) [71]	2.65		Significant radical scavenging activity against DPPH	
	DPPH free radical scavenging	**103** (μM) [70]	16.4	4.9	Significant antioxidant activity against DPPH radicals	
	SH-SY5Y	**191** (μg/mL) [41]	6.25			Weak activity
Inhibit electron transfer activity	Photosynthetic electron transport in spinach	**88**–**89** (μM) [64]	29.1 ± 6.5/22.8 ± 8.8		Strong activities	
Anti-inflammatory activity	NF-kB, TGF-β, TNF-α, IL-6, IL-8 and COX-2; caspase-3, caspase-8, Bcl-2 and Bax	**108** [66]			Potential candidate for the therapy of different vascular inflammatory diseases	
	Breast cancer cells, colon cancer cells	**177** [65]			Potential candidate for inhibiting cancer cell proliferation	

## Data Availability

Not applicable.

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
