# Peer review of "The Genus Cladosporium: A Prospective Producer of Natural Products"

_ijms, 2024, doi:10.3390/ijms25031652_

Round 1
Reviewer 1 Report
Comments and Suggestions for Authors
Please see attachment,

Author Response
Thanks very much for your kind comments, we had revised our manuscript carefully according to your comments point to point. The changes in our new version of manuscript were highlighted in red.
1. Question: Consider adding and/or replacing table 1 with a figure that indicates a series of the main findings with respect to a time line.
Answer: Thanks very much for your kind suggestion. A figure that indicates a series of the main findings with respect to a time line has been supplied in figure 21.
2. Question: It is advisable to add a small section on the structure-activity relationship.
Answer: Thanks very much for your kind comment. The structure-activity relationship has been supplied in lines 744-766.
3. Question: Add a paragraph on the possible ecological roles of these compounds produced by the Genus Cladosporium.
Answer: Thanks very much for your kind remind. The possible ecological roles of these compounds have been supplied in lines 797-806.
4.Question: Why they have great chemical diversity in the ocean?
Answer: Thanks very much for your kind comment. The reasons for the great chemical diversity of bacteria in the ocean have been supplied in lines 78-86.
5. Question: - Line 38-39: Specify which types of “cancer” and which “viral diseases”.
Answer: Thanks very much for your kind remind. Specific examples have been given of cancer and viral diseases in line 72.
6. Question: - Line 49, 122, 197, 220, 284, 287, 301, 437, 450, 459, 465, 520, 531, 591, 614: Specify the genus and/or species.
Answer: Thanks very much for your kind remind. Specific modifications are as follows.
Line 49: The specific species have been supplied in lines 43-48.
Lines 122, 437: The specific specie “the gorgonian Anthogorgia ochracea (GXWZ-07) derived” has been supplied in lines 144-145, 467-468.
Line 197: The specific specie “Chinese rice locust Oxya chinensis” has been supplied in line 221.
Line 220: The specific specie “cotton bollworm larvae Helicoverpa armigera Hubner” has been supplied in line 242.
Line 284: The soft coral has been supplied as “an unidentified soft coral” in line 309.
Line 287: The specific specie “the mangrove plant Excoecaria agallocha derived fungus Cladosporium sp. OUCMDZ-302” has been supplied in line 313.
Line 301: The specific specie “a cyanobacterial (Synechococcus elongatus, strain PCO6301) model” has been supplied in line 328.
Lines 450, 531, 614: The specific specie “sponge Callyspongia sp.” has been supplied in lines 480, 562, 648.
Line 459: The sponge has been supplied as “an unidentified Indonesian marine sponge-derived” has been in line 489.
Line 465: The specific specie “tree (Ceriops tagal) root” has been supplied in line 495.
Line 520: The mangroves has been supplied as “an unidentified mangrove” in line 551.
Line 591: The specific specie “fungal infections, such as Staphylococcus aureus, Canidia albicans, Ralstonia solanacearum and nematode Panagrellus redivivus infestations” has been supplied in lines 623-624.
7. Question: - Line 251: It is suggested to eliminate the word "modestly" and specify quantitatively.
Answer: Thanks very much for your kind remind. The word "modestly" has been deleted, and the MIC values for compound 64 have been supplied in line 270.
8. Question: - Line 281-282: Specify what types of tumors and viruses.
Answer: Thanks very much for your kind remind. The types of tumors and viruses have been supplied in lines 304-309.
9. Question: - Line 348: Specify which diseases.
Answer: Thanks very much for your kind remind. The disease has been supplied in lines 377-378.
10. Question: - Line 368: Correct "brine shrine" for brine shrimp.
Answer: Thanks very much for your kind remind. The "brine shrine" have been corrected to brine shrimp in line 398.
11. Question: - Line 385 and 386: Specify what types of cancer and skin diseases.
Answer: Thanks very much for your kind remind. The cancer and skin diseases have been supplied in lines 414-416.
12. Question: - Table 4 Page 37: Replace the word "excellent".
Answer: Thanks very much for your kind remind. The word "excellent" has been replaced with strong in table 4.
13. Question: - Improve the resolution of the figures: Figure 4, 21, 22.
Answer: Thanks very much for your kind remind. The graphic resolution for Figures 4, 21, 22 and 23 has been increased from 300 to 600.
Reviewer 2 Report
Comments and Suggestions for Authors
Fungi are recognized for their widespread presence and as one of the most productive sources of natural compounds for application. The genus Cladosporium stands out as one of the most widespread in various ecological niches. Its representatives are able to produce a wide range of secondary metabolites useful for human health, agriculture, and industry. The manuscript presented here reviews the scientific findings on the distribution, bioactivity, biosynthesis, and structural characteristics of compounds isolated from Cladosporium during the last 20 years (2000-2022). The manuscript will be of interest to the journal’s readers and could contribute to completing the information in this scientific field. The manuscript is well organized and written in good scientific style. A large number of the cited references are from the last 5 years. They are appropriate and adequate to the aim. However, the manuscript needs more attention.
Reviewer comments
1. P 2, L 69 The authors write: “Until now, no review has summarized the secondary metabolites of the genus Cladosporium.” However, there are published similar reviews, for example:
Salvatore, M.M.; Andolfi, A.; Nicoletti, R. The Genus Cladosporium: A Rich Source of Diverse and Bioactive Natural Compounds. Molecules 2021, 26, 3959. https://doi.org/10.3390/ molecules26133959
2. The aim of the review is “to explore on the secondary metabolites of genus Cladosporium, including their chemical structures and biological activities, as well as their biosynthetic pathways”. But, in the sections Abstract and Introduction (1st paragraph) the authors focus on the marine ecosystems and marine organisms. This focus does not correspond to the title and the content of the text. By reading the Abstract and the first part of the Introduction, the reader expects only marine strains to be discussed. In fact, the manuscript examines representatives of the genus Cladosporium isolated from different ecological niches.
I recommend: 1. The abstract should comply with the tendency highlighted in the text to study strains from different habitats.
2. The first paragraph of the introduction to be moved further down.
Author Response
Thanks very much for your kind comments, we had revised our manuscript carefully according to your comments point to point. The changes in our new version of manuscript were highlighted in red.
1. Question: - P 2, L 69 The authors write: “Until now, no review has summarized the secondary metabolites of the genus Cladosporium.” However, there are published similar reviews, for example: Salvatore, M.M.; Andolfi, A.; Nicoletti, R. The Genus Cladosporium: A Rich Source of Diverse and Bioactive Natural Compounds. Molecules 2021, 26, 3959. https://doi.org/10.3390/ molecules26133959.
Answer: Thanks very much for your kind comment. The sentense has been revised into "Although the secondary metabolites of Cladosporium have been reviewed[29–31], the analysis of possible biosynthetic pathways, multi-dimensional comparative analysis of natural products from marine and terrestrial sources, structure-activity relationships, and activity-based molecular network analysis have not been conducted. This review through these analysis methods to explore research on the secondary metabolites, especially the differences between marine and terrestrial derived compounds, of genus Cladosporium, including their chemical structures and biological activities, as well as their biosynthetic pathways." Through these analysis methods, active ingredients in natural products can be discovered, and the interaction relationship between them and the targets in the organism can be revealed, providing strong support for the discovery and development of new drugs.
2. Question: - The aim of the review is “to explore on the secondary metabolites of genus Cladosporium, including their chemical structures and biological activities, as well as their biosynthetic pathways”. But, in the sections Abstract and Introduction (1st paragraph) the authors focus on the marine ecosystems and marine organisms. This focus does not correspond to the title and the content of the text. By reading the Abstract and the first part of the Introduction, the reader expects only marine strains to be discussed. In fact, the manuscript examines representatives of the genus Cladosporium isolated from different ecological niches.
Answer: Thanks very much for your kind comment. The contents of the abstract and introduction have been revised to avoid misleading.
3. Question: - The abstract should comply with the tendency highlighted in the text to study strains from different habitats.
Answer: Thanks very much for your kind suggestion. The tendency of the habitats of the strains has been described.
4.Question: - The first paragraph of the introduction to be moved further down.
Answer: Thanks very much for your kind suggestion. The first paragraph of the introduction has been moved to paragraph 3 and has been revised.
Reviewer 3 Report
Comments and Suggestions for Authors
Dear Authors,
The Genus Cladosporium: A Prospective Producer of Natural Products is a very well collected compendum.
Natural products are the drug discovery leads, novel therapeutics and metabolites.
The references are uptodate, the chemdraw figures of the natural product sources are uptodate. This review is very broad. It covered terpenoids, alkaloids, aromatics, heterocyclic and aliphatic natural products. These are biologically relevant and can be a source for drug discovery efforts. Fig. 21 C. provides are clear depiction of the key biological significance. It will be more readable if this figure is enlarged with increase in the font of the each pharmacology.
I enjoyed reading it.
All the best.
Dr. Pashikanti.
Author Response
Thanks very much for your kind comments, we had revised our manuscript carefully according to your comments point to point. The changes in our new version of manuscript were highlighted in red.
Question: - Fig. 21 C. provides are clear depiction of the key biological significance. It will be more readable if this figure is enlarged with increase in the font of the each pharmacology.
Answer: Thanks very much for your kind remind. Because the content in the figure is crowded, the font can no longer be larger, the resolution of the figure has been increased from 300 to 600, and the whole figure has been enlarged.